# Keypoint-Augmented Self-Supervised Learning for Medical Image Segmentation with Limited Annotation

**Zhangsihao Yang**[*]
Arizona State University
zshyang1106@gmail.com

**Mengwei Ren**[*]
New York University
mengwei.ren@nyu.edu

**Kaize Ding**
Northwestern University
kaize.ding@northwestern.edu

**Guido Gerig**
New York University
gerig@nyu.edu

**Yalin Wang**
Arizona State University
ylwang@asu.edu

## Abstract

Pretraining CNN models (i.e., UNet) through self-supervision has become a powerful approach to facilitate medical image segmentation under low annotation regimes. Recent contrastive learning methods encourage similar global representations when the *same* image undergoes different transformations, or enforce invariance across *different* image/patch features that are intrinsically correlated. However, CNN-extracted global and local features are limited in capturing long-range spatial dependencies that are essential in biological anatomy. To this end, we present a keypoint-augmented fusion layer that extracts representations preserving both short- and long-range self-attention. In particular, we augment the CNN feature map at multiple scales by incorporating an additional input that learns long-range spatial self-attention among localized keypoint features. Further, we introduce both global and local self-supervised pretraining for the framework. At the global scale, we obtain global representations from both the bottleneck of the UNet, and by aggregating multiscale keypoint features. These global features are subsequently regularized through image-level contrastive objectives. At the local scale, we define a distance-based criterion to first establish correspondences among keypoints and encourage similarity between their features. Through extensive experiments on both MRI and CT segmentation tasks, we demonstrate the architectural advantages of our proposed method in comparison to both CNN and Transformer-based UNets, when all architectures are trained with randomly initialized weights. With our proposed pretraining strategy, our method further outperforms existing SSL methods by producing more robust self-attention and achieving state-of-the-art segmentation results. The code is available at `https://github.com/zshyang/kaf.git`.

## 1 Introduction

Large-scale and diverse source of training data has significantly empowered the generalization ability of supervised segmentation models [52, 77, 34]. However, in biomedical image analysis, manual delineation of images/volumes requires extensive domain knowledge and can be extremely time-consuming and error-prone. Recently, self-supervised learning (SSL) [9, 25, 18, 45, 7, 72, 51] has emerged in medical vision to pretrain a segmentation backbone (e.g. UNet) leveraging large scale unlabelled data, to serve as a better initialization before finetuning the model with limited annotation.

---

[*]Equal contribution.

37th Conference on Neural Information Processing Systems (NeurIPS 2023).

As one of the most successful learning paradigms in SSL, contrastive learning pulls closer the representations of positive pairs (i.e. the same image under different augmentations), while pushing apart the negative pairs (i.e. different images) [9, 28, 10, 25, 46, 11, 12]. However, in medical images, strong anatomical similarities often exist across different images, resulting in an increased number of false negative samples which is detrimental to the representation learning process. Therefore, heuristics about relationship across different images are utilized to constitute positive pairs, such as encouraging the global representations of 2D slices at similar volumetric positions [7, 72] to be close. To further benefit pixel-level tasks such as segmentation, local representation learning methods [7, 51] learn distinctive local representations by attracting patch features at the same position from two augmented views [7], or across registered intra-subject volumes [51]. Nevertheless, the potential dependencies among patch features that are spatially distant are not properly taken into account. Such region-awareness can be enhanced by attracting spatial features points with the same semantics [35, 27, 38, 75, 1]. However, their training requires segmentation ground truth, which limits their applicability in medical vision when adequate annotation is unavailable.

In this paper, we incorporate the long-range spatial dependencies into the UNet backbone with a plug-and-play keypoint-augmented fusion layer (KAF layer), accompanied by keypoint-enhanced global and local objectives for pretraining the network with self-supervision. To do so, we first identify a set of sparse keypoint locations from the input image. On the output feature map of each UNet encoder block, the keypoint features are sampled from the convolutional feature map, and the attention among them are modeled through a Vision Transformer. The output is then scattered back to the feature grid, and gets fused to the original CNN feature, to explicitly provide clues on the spatial long-range dependencies. Further, we propose the global and local self-supervised learning objectives to pretrain the keypoint-augmented fusion layer -enhanced network. The global contrastive loss is applied on both the UNet bottleneck feature, as well as keypoint-augmented global feature obtained by aggregating the multiscale keypoint features. To further benefit pixel-level down-stream tasks, we identify the correspondence among keypoints across different image slices and maximize their feature similarities with a local self-supervised objective.

Our contributions are threefold. (1) We develop a plug-and-play keypoint-augmented fusion layer that augments the convolutional feature map with long-range dependencies among keypoint features. When trained with only limited annotation, our proposed KAF layer achieves noticeably better results than the CNN-only and/or Transformer-based backbone. (2) We further propose the keypoint-aware global and local self-supervised learning objectives to pretrain our model before finetuning it with annotations. (3) We conduct extensive experiments on three MRI and CT datasets, and achieve state-of-the-art few-shot segmentation results, verifying both architecture advantage of the proposed layer and our pretraining strategies.

## 2   Related Work

**Biomedical Image Segmentation** is the process of dividing images into various segments or regions and isolating certain structures of interest in modalities such as magnetic resonance imaging (MRI) and computed tomography (CT) [50, 37]. UNet [52] is the pioneering method for addressing biomedical image segmentation tasks using deep learning. Subsequent research aims to enhance or apply the UNet model in various ways, such as extending to 3D volumes [44, 64], auto-configuration [29], altering skip connections [77, 66], and replacing and/or combining the convolutional backbone layers with Transformers [23, 5, 22, 30]. However, a common limitation shared by all end-to-end methods lies in the requirement of large-scale and high-quality biomedical annotations which are both time-consuming and requires expert knowledge [52]. This limitation underscores the need for novel methods that can learn effectively with fewer or without annotations.

**Self-Supervised Learning (SSL)** learns meaningful representations from unlabeled datasets with self-supervised objectives, and the pretrained network constitutes a better initialization than random initialization when it is finetuned on downstream tasks with limited label supervision, e.g., classification or segmentation. Existing SSL algorithms either define pretext tasks with heuristics, or employ self-supervised objectives to learn invariance and/or equivariance of image features. Early pre-text tasks based pretraining designs objective that does not require additional labels, such as rotation prediction [18], context restoration [48], reconstruction [32], among others [24, 42, 15]. Recently, contrastive learning has become a prevailing paradigm in SSL and has shown great success in both natural image [6, 59, 78, 63, 76] and medical vision [7, 72, 51, 58, 71, 70, 56, 49, 8, 62, 16, 47, 73, 68].

It enforces similar representations between positive pairs and distinct representations between negative pairs. Typically, similarity is determined in an unsupervised way, i.e., the same image under different transformations are recognized as positive pairs, whose global representations should remain invariant, and features from different images are pulled apart. In medical images, heuristics about relationships among different images are also utilized to define positives, e.g., 2D slices at similar volumetric positions [7, 72], or local patches at the same location from intra-subject volumes [51].

**Long-Range Dependency** generally cannot be well captured in CNN due to its locality nature, thus neglecting the potential relationship across spatially distant local regions that possess high semantic consistency. To tackle the limitation, existing works focus on either (1) architectural design that injects long-range attention, or (2) SSL objectives that explicitly enforce regional similarity. For the first line of research, a series of non-local networks have been proposed. For instance, Vision Transformer (ViT) [14] enables self-attention on uniformly sampled image patches and has been incorporated into CNN backbones for segmentation tasks [22, 5, 36, 26]; [54, 43, 20, 65] adopt graph networks to model dependencies among context, keypoints or boundaries. The other line of research captures regional correlation with customized contrastive formulation objective, typically in a label-supervised manner [35, 27, 75, 1, 38], where local features within the same semantic label form the positive pairs. In our setting, we eliminate the need of segmentation labels via establishing correspondence among detected keypoints and apply local self-supervision on the keypoint features.

**Keypoint Descriptors** have been successfully exploited in various domains to improve the performance on tasks such as pose estimation [19, 4], image generation [57] and object detection [79, 17]. Specifically, in biomedical imaging, keypoint descriptors have found applications in a variety of areas, including anatomy object classification [31], medical image retrieval [74], segmentation tasks [69, 61], motion estimation of organs [53], and medical image registrations [21, 3]. They aim to incorporate sparse and localized key points into the training process, in order to attend the network to the most important features within the image. This is typically done via first localizing a sparse set of keypoint locations in the image, where keypoint features are obtained and their interactions are modeled. Finally, the feature is aggregated and used for downstream task prediction. In this work, we propose to inject learnable keypoint descriptors into the UNet building block so that the network is able to model complex long-range attention without a significant increase in computational cost.

## 3  Methodology

Fig. 1 illustrates an overview of the proposed method. Built upon a segmentation backbone (i.e., UNet), we attach our proposed keypoint-augmented fusion layer (Sec. 3.1) after each convolutional block, to inject long-range self-attention into the original convolutional feature map. Further, we propose both global and local self-supervision (Sec. 3.2) to pretrain the network before finetuning with limited annotation.

### 3.1  Keypoint-Augmented Fusion (KAF) Layer

The proposed KAF layer is diagrammed in the red dashed box in Fig. 1. Given an input image $\mathcal{X} \in \mathbb{R}^{W \times H \times C}$, we start with detecting keypoints $\mathcal{K} \in \mathbb{R}^{N \times 2}$ using a keypoint detector $\mathcal{D}_k : \mathbb{R}^{W \times H \times C} \to \mathbb{R}^{N \times 2}$. In particular, we employ the Scale-Invariant Feature Transform (SIFT) [41]. We note that alternative keypoint detection methods can also be applied, and we have included the ablation in the appendix. At $l$-th layer of the 2D UNet, we acquire $N$ CNN features $\mathcal{F}_k^l \in \mathbb{R}^{N \times C^l}$ from the dense convolutional feature map $\mathcal{F}^l \in \mathbb{R}^{W^l \times H^l \times C^l}$, based on the keypoint coordinates $\mathcal{K}^l$. Note that at each scale, $\mathcal{K}^l$ is acquired by re-scaling $\mathcal{K}$ with the resolution ratio between the input image and the feature map. Then, we map the keypoint features to the embedding space with dimension $E \in \mathbb{R}$ with a projector $\phi : \mathbb{R}^{N \times C^l} \to \mathbb{R}^{N \times E}$. We use a single-layer MLP in our implementation as the projector. The embedded features are then fed into a Transformer $\mathcal{T} : \mathbb{R}^{N \times E} \to \mathbb{R}^{N \times E}$ to learn the self-attention among keypoints. and we define the transformed feature as $U^l \in \mathbb{R}^{N \times E}$ (which is further used as the keypoint feature at layer $l$ for calculating the SSL losses). Additionally, to properly propagate the keypoint features learned at the current scale $l$ to $l+1$ within the convolutional UNet, $U^l$ is scattered back into the feature grid indexed by the rescaled keypoint position $\mathcal{K}^l$. The output is sparse feature map $\mathcal{F}_s^l$ which retains information exclusively at the keypoint positions, and the rest of the feature map are assigned with zeros.

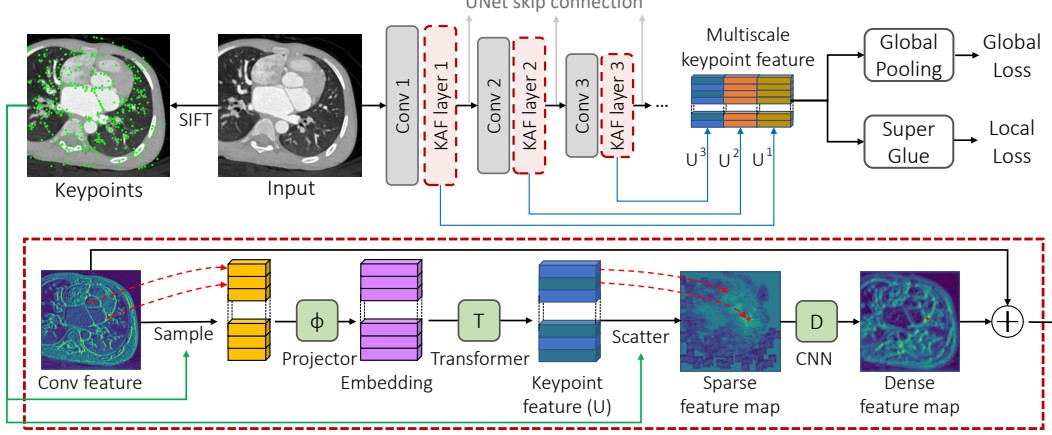

Figure 1: **Top**: The overview of the proposed SSL framework that incorporates both local and global self-supervision with the keypoint guidance. The UNet decoder is omitted for better readability. **Bottom**: The proposed Keypoint-Augmented Fusion Layer (KAF layer), which learns long-range spatial dependencies among localized keypoint features. We insert the KAF layer after each encoder block of the UNet backbone to augment the original convolutional features.

We further attach a two-layer CNN $\mathcal{D}_l$ to diffuse the sparse keypoint feature map into a dense image feature. Lastly, we concatenate the diffused keypoint feature map with the input feature map $F^l$, constituting the final output of the proposed layer, represented as $\mathcal{F}_o^l = \texttt{Concat}(F^l, \mathcal{D}_l(\mathcal{F}_s^l))$.

## 3.2 Keypoint-Augmented Self-Supervised Learning

Furthermore, we design the global and local self-supervised learning algorithms to enhance the keypoint-augmented feature learning, thus the learned feature representations can be well generalized to different downstream tasks through finetuning.

**Global SSL Loss.** In order to apply a global contrastive loss (e.g., [9, 72]) that learns image wise feature similarity, we first aggregate multiscale keypoint features and extract a keypoint-enhanced global representation for each input image. Specifically, keypoint features from four encoder blocks are first concatenated as $Concat(U^1, U^2, U^3, U^4)$, which is then fed into a Multilayer Perceptron (MLP) to get a global feature $g_i$, where $i$ is the index of the input image from the dataset. In our training, we adopt the global contrastive loss from PCL [72] on the keypoint-enhanced global feature, which assumes 2D slices within a certain positional distance (within the 3D volume) threshold to be anatomically similar and constitute similar pairs. Formally, the global self-supervised learning loss function can be formulated as:

$$L_{global}^i = -\frac{1}{|\Omega_i^+|} \sum_{j \in \Omega_i^+} \log \frac{e^{\frac{sim(g_i, g_j)}{\tau}}}{\sum_{k=1}^{2N} 1_{i \neq k} \cdot e^{\frac{sim(g_i, g_k)}{\tau}}}, \tag{1}$$

where $sim(\cdot, \cdot)$ is the cosine similarity between two vectors in the embedding space, $\tau$ indicates the temperature term. $\Omega^+$ is the set of positive images to the input $x_i$.

We additionally keep the global contrastive loss applied on the features from the last layer of the UNet encoder, as commonly done in existing SSL literature [72, 7, 9]. To do so, a global pooling layer transforms the feature $\mathcal{F}^{last}$ from $\mathbb{R}^{W^{last} \times H^{last} \times C^{last}}$ to $\mathbb{R}^{1 \times C^{last}}$. Afterward, an MLP projector is applied to obtain the global embedding for contrastive objectives, denoted as $c_i$. Similar to Eq. 1, the loss can be formulated by substituting $g_i$ with $c_i$ and $g_j$ with $c_j$ in Eq. 1. As we adopt PCL loss [72] to the global CNN feature, we denote this loss as $L_{PCL}^i$ in the following context.

**Local SSL Loss.** While [72] only considers image-wise similarity based on the slice positions, we claim that pixel-wise similarity across slices should also be exploited to supervise the keypoint feature learning, enhancing fine-grained and localized control of the learned representation. We

leverage the intrinsic structural correlation in 3D medical imaging, where nearby slices display strong local similarities. For instance, in cardiac imaging, anatomical structures such as blood vessels and ventricles span across multiple 2D slices within the 3D volume. Therefore, keypoints within a certain spatial distance from adjacent slices are likely to be semantically correlated and form positive pairs.

In order to locate the correspondence for each keypoint, we define two major criteria based on heuristics: (1) the spatial distance in the 2D plane between two correspondences should be within a threshold; (2) the distance between their SIFT features should be similar. Correspondingly, we first sample two positive slices along with their keypoints $(\mathcal{X}_i, \mathcal{K}_i)$ and $(\mathcal{X}_j, \mathcal{K}_j)$, where the positional distance between $\mathcal{X}_i$ and $\mathcal{X}_j$ fall within the positive threshold defined in [72]. For each keypoint $a \in \mathcal{X}_i$, we aim to find its correspondence from $\mathcal{X}_j$. We do so by first filtering out a set of keypoints whose Manhattan distances with $a$ is less than a predefined threshold. Then, we find the keypoint $b$ as the correspondence of $a$ based on the closest L2 distance of the SIFT features. Once matched, $(a, b)$ is added into the groundtruth match set $\mathcal{M} \subset \mathcal{A} \times \mathcal{B}$. However, if no points in slice $\mathcal{X}_j$ meet the criteria, the candidate point $a$ is treated as a negative sample and added to the unmatched set $\mathcal{I} \subseteq \mathcal{A}$. Similarly, for slice $\mathcal{X}_j$, the candidate point without match is added to the unmatched set $\mathcal{J} \subseteq \mathcal{B}$. $\mathcal{M}, \mathcal{I}, \mathcal{J}$ are then used as the groundtruth for the keypoint feature matching detailed below.

We repurpose SuperGlue [54] as the backbone method to learn the correspondence between keypoint features on slice $\mathcal{X}_i$ and keypoints on slice $\mathcal{X}_j$, supervised by the correspondence extracted above. We first compute the feature associated with keypoints $U_i = \mathtt{Concat}(U_i^1, U_i^2, U_i^3, U_i^4)$, and $U_j = \mathtt{Concat}(U_j^1, U_j^2, U_j^3, U_j^4)$, along with their groundtruth matches $\mathcal{M}$ and unmatched sets $\mathcal{I}$ and $\mathcal{J}$. Then, SuperGlue takes $U_i, U_j, \mathcal{K}_i$, and $\mathcal{K}_j$ as inputs and computes an output $\bar{P} \in \mathbb{R}^{(N_i+1) \times (N_j+1)}$ through self- and cross- graph message passing. Here, $\bar{P}_{a,b}$ represents the probability that keypoint $a \in \mathcal{K}_i$ matches with $b \in \mathcal{K}_j$. Given $\mathcal{M}, \mathcal{I}$, and $\mathcal{J}$, we minimize the negative log-likelihood of the assignment $\bar{P}$, formulated as follows:

$$L_{local}^i = - \sum_{(a,b) \in M} \log \bar{P}{a,b} - \sum_{a \in I} \log \bar{P}_{a, N_i+1} - \sum_{b \in J} \log \bar{P}_{N_j+1, b}. \tag{2}$$

By establishing such localized correlation among keypoint features, we assume the network is able to better recognize local correspondence even when the input image undergoes various transformations. Such localized correlation ensures that the network can maintain a consistent understanding of the relationships between keypoint features, regardless of changes in the image appearance or orientation. Our assumption is empirically verified from the computed equivariant self-attention map shown in Fig. 3.

**Total SSL Loss.** With the proposed keypoint-augmented fusion layer integrated into existing CNN backbone, the model can be pretrained with the above global and local losses, optionally with existing global contrastive loss (in our case, PCL [72]). The final objective is:

$$L_{total} = w_1 \cdot L_{PCL} + w_2 \cdot L_{global} + w_3 \cdot L_{local}, \tag{3}$$

where the coefficients $w_1$, $w_2$, and $w_3$ are employed to balance the contributions of each term. In our experiments, we use a grid search method to optimize these coefficients.

## 4 Experiments

**Datasets.** We conduct experiments on two publicly accessible cardiac MRI datasets for the task of segmentation under limited annotation. For fair benchmarking, we follow the same preprocessing steps as [72]. For each cross-validation fold, $N$ images are held-out for validation, and varying $M$ images from the remaining images are used for the few-shot training. We additionally include results on a non-cardiac CT dataset Synapse[1] in the appendix to illustrate the generalization of our method.

CHD (Congenital Heart Disease) [67] is a CT dataset consisting of 68 3D cardiac images, covering patent ages ranging from 1 month to 21 year. It includes 14 types of congenital heart disease and the segmentation labels consists of seven distinct substructures: left ventricle (LV), right ventricle (RV), left atrium (LA), right atrium (RA), myocardium (Myo), aorta (Ao), and pulmonary artery (PA). For each cross-validation fold, $N$=18 images are used for validation, and $M$ images are selected from the remaining 50 images for few-shot training.

---

[1] `https://www.synapse.org/#!Synapse:syn3193805/wiki/217789`

Table 1: Benchmark results on CHD and ACDC datasets under both random initialized weights, and pretrained weights from SSL. $M$ is the number of patients used in supervised training. We perform 5-fold cross-validation and the mean (standard deviation) dice scores are reported.

| | | CHD (68 patients in total) | | | | | | |
|---|---|---|---|---|---|---|---|---|
| Init. | Method | $M$=2 | $M$=6 | $M$=10 | $M$=15 | $M$=20 | $M$=30 | $M$=51 |
| Random | UNet [52] | 0.184(.06) | 0.508(.06) | 0.584(.05) | 0.627(.05) | 0.658(.04) | 0.693(.04) | 0.754(.02) |
| | Swin-Unet [5] | 0.291(.07) | 0.543(.07) | 0.624(.04) | 0.675(.05) | **0.717(.04)** | **0.732(.05)** | 0.784(.03) |
| | SwinUNETR [58] | 0.345(.07) | 0.565(.06) | 0.638(.05) | 0.682(.06) | 0.711(.05) | 0.725(.06) | **0.785(.03)** |
| | Ours | **0.344(.05)** | **0.576(.07)** | **0.646(.03)** | **0.686(.03)** | 0.706(.03) | 0.728(.04) | 0.778(.03) |
| SSL pretrain | Rotation [18] | 0.171(.06) | 0.488(.07) | 0.575(.04) | 0.625(.04) | 0.651(.04) | 0.691(.04) | 0.749(.03) |
| | PIRL [45] | 0.196(.07) | 0.504(.08) | 0.617(.05) | 0.658(.05) | 0.674(.04) | 0.714(.04) | 0.761(.03) |
| | SimCLR [9] | 0.192(.06) | 0.515(.06) | 0.599(.06) | 0.631(.05) | 0.666(.05) | 0.699(.05) | 0.756(.03) |
| | GLCL-*global* [7] | 0.255(.10) | 0.564(.04) | 0.646(.03) | 0.669(.04) | 0.697(.04) | 0.725(.04) | 0.766(.03) |
| | GLCL-*full* [7] | 0.286(.06) | 0.555(.07) | 0.614(.06) | 0.666(.04) | 0.694(.04) | 0.733(.04) | 0.772(.03) |
| | CAiD [56] | 0.265(.08) | 0.581(.06) | 0.647(.04) | 0.684(.04) | 0.700(.04) | 0.737(.04) | 0.771(.02) |
| | PCL [72] | 0.356(.08) | 0.600(.06) | 0.661(.05) | 0.686(.05) | 0.716(.04) | 0.735(.05) | 0.774(.03) |
| | Ours | **0.392(.06)** | **0.636(.06)** | **0.693(.03)** | **0.712(.03)** | **0.728(.04)** | **0.754(.04)** | **0.788(.03)** |

| | | ACDC (100 patients in total) | | | | | | |
|---|---|---|---|---|---|---|---|---|
| Init. | Method | $M$=2 | $M$=6 | $M$=10 | $M$=15 | $M$=20 | $M$=30 | $M$=80 |
| Random | UNet [52] | 0.588(.07) | 0.782(.03) | 0.840(.03) | 0.876(.01) | 0.894(.01) | 0.909(.01) | **0.928(.00)** |
| | Swin-Unet [5] | 0.181(.07) | 0.483(.09) | 0.610(.05) | 0.700(.04) | 0.735(.02) | 0.772(.01) | 0.870(.01) |
| | SwinUNETR [58] | 0.567(.04) | 0.740(.07) | 0.799(.05) | 0.850(.02) | 0.881(.01) | 0.904(.01) | 0.922(.00) |
| | Ours | **0.655(.05)** | **0.827(.05)** | **0.871(.02)** | **0.897(.01)** | **0.901(.01)** | **0.915(.00)** | 0.927(.00) |
| SSL pretrain | Rotation [18] | 0.572(.08) | 0.809(.03) | 0.868(.02) | 0.886(.01) | 0.898(.01) | 0.910(.01) | 0.925(.00) |
| | PIRL [45] | 0.492(.03) | 0.823(.04) | 0.865(.01) | 0.880(.02) | 0.896(.02) | 0.912(.01) | 0.927(.00) |
| | SimCLR [9] | 0.352(.06) | 0.725(.08) | 0.824(.04) | 0.869(.02) | 0.894(.01) | 0.913(.01) | 0.927(.00) |
| | GLCL-*global* [7] | 0.636(.05) | 0.803(.04) | 0.872(.01) | 0.891(.01) | 0.902(.01) | 0.913(.01) | 0.927(.01) |
| | GLCL-*full* [7] | 0.642(.06) | 0.802(.03) | 0.877(.01) | 0.891(.01) | 0.904(.02) | 0.912(.00) | 0.927(.00) |
| | CAiD [56] | 0.483(.11) | 0.822(.02) | 0.879(.02) | 0.896(.01) | 0.905(.00) | 0.914(.00) | 0.926(.00) |
| | PCL [72] | 0.671(.06) | 0.850(.01) | 0.885(.01) | 0.904(.01) | 0.909(.01) | 0.919(.00) | 0.929(.00) |
| | Ours | **0.741(.03)** | **0.873(.01)** | **0.895(.01)** | **0.908(.01)** | **0.915(.00)** | **0.921(.00)** | **0.930(.00)** |

ACDC (Automatic Cardiac Diagnosis Challenge) [2] is a MRI dataset consists of cardiac images from 100 patients under several different pathologies. Manual expert segmentation of the RV, LV cavities, and Myo are conducted for the volumes from end-diastolic and end-systolic phase. For each cross-validation fold, $N = 20$ images are used as the validation set, and $M$ images from the remaining 80 images are used for supervised finetuning.

**Baselines and Evaluation Settings.** We evaluate segmentation performance trained with limited annotation under both (1) random initialized network backbones (UNet v.s. Ours) to verify the architectural advantages of the proposed keypoint-augmented fusion layer. (2) pretrained weights from our self-supervision compared with various self-supervised pretraining methods including pre-text tasks [18, 45] and contrastive learning [9, 7]. We quantify the segmentation performance via the Dice coefficient with five-fold cross validataion following the set up in [72]. Additionally, we conducted ablation studies to isolate each component in the pipeline to assess individual effects.

**Implementation Details.** For network configuration, we build our framework on 2D UNet, and set the starting number of channels of the network as 32 for CHD, and 48 for ACDC. Five convolutional blocks are included in the encoder and four blocks are used within the decoder. Each block contains two convolution layers, followed by batch normalization, and ReLU activation. Each encoder block downsample the feature map by a factor of 2, while simultaneously doubling the number of channels. In between the first four encoder blocks, we further append the proposed keypoint-augmented fusion layer , and concatenate the output to the original convolutional feature map (Fig. 1). Each Transformer inside KAF layer consists of six self-attention layers, and the comparison among different number of self-attention layers are presented in ablation study below. For pretraining, we assign loss weights $w_1, w_2, w_3$ to 1.0, 1.0, and 0.01, respectively. We employ the SGD optimizer with a learning rate of 0.002 and batch sizes of 32 for CHD and ACDC. A cosine learning rate scheduler is utilized, with the minimum learning rate set to 0. We pretrain the model for 50 epochs. For finetuning, we use the standard cross-entropy loss with the Adam [33] optimizer, with learning rates of to $5 \times 10^{-5}$ for CHD, $5 \times 10^{-4}$ for ACDC. The batch size is set to 10, and we finetune on CHD for 100 epochs and on ACDC for 200 epochs. Additional implementation details are provided in the appendix.

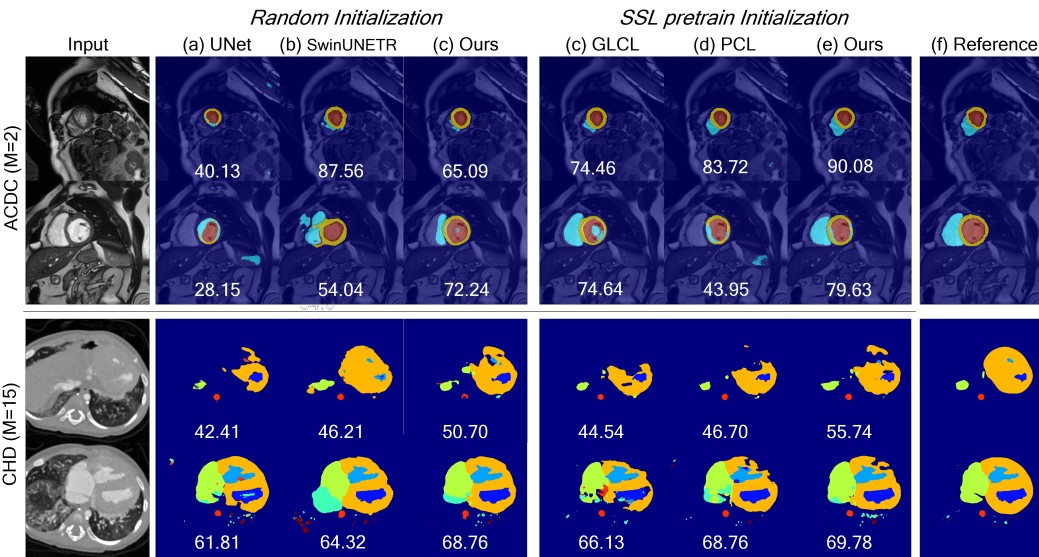

Figure 2: **Few-shot segmentation results.** Colomn (a)-(c) show a comparison among the UNet backbone, SwinUNETR [58], and our proposed backbone trained on limited dataset with random weight initialization. Our architecture allows for reduced false positive and improved segmentation accuracy. Column (c)-(e) compares our pretraining strategy with the existing state-of-the-art SSL methods [7] and [72]. When finetuned with small number of labeled dataset, our method presents higher coherence with the reference image (column (f)). The number beneath each prediction represents the dice value for the displayed slice. For better readability, we scale the value by 100.

**Segementation Results without Pretraining.** To verify the benefits of the proposed layer, we start with analyzing the performance gain without any pretraining, to isolate and evaluate the impact solely attributed to the architectural change on top of the UNet backbone. Specifically, we train the randomly initialized UNet and our backbone with varying limited number of training data, and the results are shown in Table 1 under `Init.Random`. Compared with the UNet backbone with an identical number of convolutional blocks and network configurations, introducing KAF layer consistently and significantly improves the segmentation results on both datasets across various training sample sizes. The empirical findings strongly support our assumption that incorporating long-range dependencies into the segmentation backbone is crucial and helps better modeling and utilizing regional image information when only limited annotations are available. Notably, our network demonstrates superior performance over the UNet baselines when the training size is extremely small. For instance, when the sample size is reduced to two ($M = 2$) on the CHD dataset, the inclusion of KAF layer leads to 87% performance gain in dice score over UNet. We also include Transformer-based UNets [5, 58], which also consider long-range spatial dependencies. Our observation is consistent with the conclusion in [5], where Transformers may be more severely affected by the initialization and require large $M$ in order to perform well, while method obtain much better results when only limited data is available.

Fig. 2 `Random Initialization` shows a visual comparison of the segmentation between UNet and Ours. While both methods suffer from sub-optimal generalization ability on the input image from the unseen validation set, our method largely reduces false positive prediction and presents higher similarity with the reference image on the right. Such improvement can be attributed to the utilization of keypoints, which enhances the feature learning process by directing attention to a more constrained set of regions rather than the entire image space. we deduce that this focused awareness contributes to better localization of important features thus leading to better semantic segmentation.

**Segmentation Results with Pretraining and Finetuning.** To further enhance the segmentation performance and promote better generalization of our model, we pretrain the network with the proposed SS objectives and leverage the pretrained weights as the initialization for fine-tuning on the same amount of labeled dataset. Quantitative results are shown in Table 1 under the tab `SSL pretrain initialization`, where benchmarking is conducted across existing state-of-the-art SSL pretraining methods. Notably, our pretraining technique yields significant improvements in few-shot

Table 2: Ablation study of our framework over (1) architecture design: the number of self-attention layers within the transformer (#T), scales to insert KAF layer $(l_1, \cdots, l_4)$; and (2) pretraining hyper-parameters $(w_1, w_2, w_3)$. Five-fold cross-validation results on both datasets are reported.

| Init. | Exp | Architecture design | | | | | Pretraining | | | Dice | |
|---|---|---|---|---|---|---|---|---|---|---|---|
| | | #T | $l_1$ | $l_2$ | $l_3$ | $l_4$ | $w_1$ | $w_2$ | $w_3$ | CHD ($M$=15) | ACDC ($M$=6) |
| Random | A | 9 | 0 | 0 | 0 | 0 | - | - | - | 0.627(.05) | 0.782(.03) |
| | B | 9 | 1 | 1 | 1 | 0 | - | - | - | 0.658(.04) | 0.806(.04) |
| | C | 9 | 1 | 1 | 0 | 1 | - | - | - | 0.677(.04) | 0.817(.04) |
| | D | 9 | 1 | 0 | 1 | 1 | - | - | - | 0.667(.04) | 0.814(.04) |
| | E | 9 | 0 | 1 | 1 | 1 | - | - | - | 0.666(.04) | 0.811(.04) |
| | F | 9 | 1 | 1 | 1 | 1 | - | - | - | 0.686(.03) | 0.827(.05) |
| | G | 6 | 1 | 1 | 1 | 1 | - | - | - | 0.690(.03) | 0.824(.03) |
| | H | 3 | 1 | 1 | 1 | 1 | - | - | - | 0.677(.03) | 0.814(.03) |
| SSL pretrain | I | 9 | 1 | 1 | 1 | 1 | 1 | 0 | 0 | 0.699(.04) | 0.867(.02) |
| | J | 9 | 1 | 1 | 1 | 1 | 0 | 1 | 0 | 0.701(.03) | 0.865(.02) |
| | K | 9 | 1 | 1 | 1 | 1 | 1 | 1 | 0 | 0.711(.03) | 0.865(.01) |
| | L | 9 | 1 | 1 | 1 | 1 | 1 | 1 | 0.01 | **0.712(.03)** | **0.873(.01)** |

segmentation tasks compared to the random initialization. Moreover, in comparison to alternative pretraining strategies that do not incorporate long-range dependencies, our method consistently achieves state-of-the-art dice scores across different numbers of training subjects for both datasets, demonstrating the benefits of the proposed keypoint-augmented SSL objectives. In Fig. 2 SSL `initialization`, we present visual comparisons between our method and the best-performing SSL methods GLCL [7] and PCL [72]. Visually, we observe that GLCL tends to produce false positives or incorrectly labels the anatomy, as evidenced by the CHD result. Conversely, PCL tends to generate false negatives, as observed in the ACDC example. In contrast, our proposed method achieves the highest level of similarity with the reference image, accurately capturing the desired anatomical structures. These visual comparisons further reinforce the effectiveness and superior performance of our approach in comparison to existing SSL methods.

**Ablation Study.** As our proposed framework consists of both architectural change on the UNet building blocks and pretraining strategy, we conduct ablation analysis over different architectural configurations and pretraining objectives, with results presented in Table 2. Additional ablations on keypoint detection method and threshold for identifying correspondence are provided in the appendix.

Architecture (Exp `A-H`). We start with architectural analysis by training different randomly initialized backbone networks under different configurations over the number of self-attention (#T) within the KAF layer , and the injection of KAF layer at different encoder scales $l_1, \cdots, l_4$.

Exp `A` represents a UNet backbone without the use of KAF layer . With the same #T, Exp `F` inserts four KAF layer after each of the encoder layer. A significant performance gain was obtained on CHD by 0.06, and on ACDC by 0.04. To analyze layerwise effects, we remove one KAF layer each time from Exp `B-E`, and observe a degradation in performance, indicating the importance of utilizing a multiscale setting. We observe that the last layer plays the most crucial role in achieving superior segmentation results. In addition, we investigate the effect of the number of self-attention within the Transformer used in our architecture in Exp `F-H`. We observe that the performance reaches a plateau with $\#T = 9$, and when it reduces to 3, the performance further degrades. Nevertheless, Exp `H` still overperforms Exp `A`, indicating the benefits of introducing long-range self-attention.

Pretraining Strategy (Exp `I-L`). On the architectures of Exp `F`, we further add the proposed pretraining strategies, and isolate the different loss weights during the training. $w_1, w_2$ indicate the global PCL loss applied on the convolutional feature and keypoint features respectively, $w_3$ indicates the local loss weight (See Eq. 3). Compared with single global loss on either the global CNN feature (Exp `I`) or the keypoint feature (Exp `J`), a combination of the two terms (Exp `K`) shows improved results on CHD. The best performance is achieved with a combination of both global and local loss (Exp `L`). Additional weight tuning details are provided in appendix.

**Qualitative Analysis of the Learned Self-Attention.** In Fig. 3, we qualitatively compare the learned self-similarity from our pretraining, with the established PCL [72] (global), and GLCL [7] (global and local) pretraining. Given an input image $x$, we perform two random transformations $T_1$ and $T_2$ to obtain a simulated positive pairs in contrastive set up. A query point is randomly sampled from

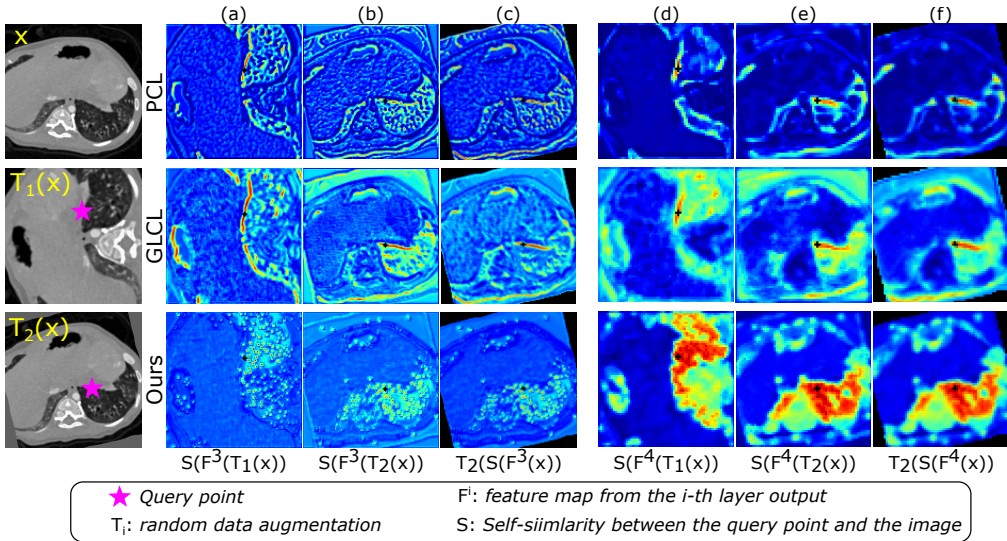

Figure 3: Learned multiscale self-similarity between the query point feature (star) and all other feature points within the same feature map from the UNet encoder. The comparison between (a) and (b), (d) and (e) indicates that our method maintains better invariance of the local feature self-similarity under different transformations. Additionally, the coherence between (b) and (c), (e) and (f) verify that our method learns better equivariance. Details are elaborated in Sec. 4.

the $x$ and gets transformed to $T_1(x)$ and $T_2(x)$ respectively (stars in Fig. 3). The image is fed into the pretrained network, and the feature map from the $i$-th layer is obtained. The self-similairty $S$ is computed as the dot product between the query point feature and other feature points within the same map. Fig. 3 visualizes the self-attention from the third and fourth layer of the pretrained UNet ($l_3, l_4$).

We observe two major differences between our method and PCL and GLCL: (1) our keypoint-augmented fusion layer helps the network to concentrate its attention on more constrained sub-regions. For example, both PCL and GLCL tend to learn high response values between the query point and background regions (e.g., Fig. 3 (b), (e)), whereas our method exhibits a more refined attention mechanism, directing the focus of the query point towards anatomically relevant regions, which helps the model to capture and leverage important features within the desired regions, leading to improved segmentation performance. (2) our proposed self-supervision helps maintain better local equivariance of the self-attention, i.e., with the same query point location, its interaction with other points remains identical no matter how the image is transformed. This is verified by a comparison between Fig. 3 (a) v.s. (b), and (d) v.s. (e), where our method achieves the best consistency of the local similarity when features are extracted from the image with different transforms.

Additionally, we calculate the self-attention based on the input image $x$ and transform the resulting similarity map using $T_2$. The transformed similarity map is expected to exhibit a high level of coherence with the similarity map obtained from the transformed input. Compared with PCL and GLCL, our method demonstrates stronger consistency between (b) and (c), as well as (e) and (f), which validates the advantage of introducing the keypoint-augmented self-supervised learning.

## 5  Discussion

In this work, we present a keypoint-augmented fusion layer to incorporate long-range dependencies into the UNet-based segmentation framework, accompanied by global & local SSL objectives for pretraining. Open questions exist and will be included in future work. Currently, our backbone is built upon 2D UNet following [72, 7], while 3D UNet naturally serves as a better baseline in many biomedical segmentation tasks [51]. Although our method is generic and can be scaled up to 3D data, it may require different architectural configurations to accommodate the increasing number of keypoints within the 3D volume. Besides, our assumption for identifying correspondence may fail in 3D volume with the sparse acquisition, when the neighboring slices no longer maintain semantically

similar structures. Therefore, more robust and complicated criteria for obtaining the correspondence is required.

## Acknowledgements

Co-authors Zhangsihao Yang and Yalin Wang are grateful to R21AG065942, R01EY032125, and R01DE030286.

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

## A  Additional Experiments

**More comparison with Transformers** As we pointed out in the main text, Transformer-based architecture may be extremely sensitive to weight initialization, and may benefit from effective pretraining. Therefore, we further pretrain the SwinUNETR with two SSL strategies: PCL [72] and the self-supervised loss proposed in [22] (in Tabel 3). Both of these pretraining methods led to observable improvements over random initialization, while our proposed method still achieved the best performance.

Table 3: Comparison with Transformer-based methods on CHD and ACDC datasets under both random initialized weights, and pretrained weights from SSL. $M$ is the number of patients used in supervised training. We perform 5-fold cross-validation and the mean (standard deviation) dice scores are reported.

| Init. | Method | CHD (68 patients in total) | | | | | | |
|---|---|---|---|---|---|---|---|---|
| | | $M$=2 | $M$=6 | $M$=10 | $M$=15 | $M$=20 | $M$=30 | $M$=51 |
| Random | SwinUNETR-3d [58] | 0.040(.01) | 0.189(.03) | 0.332(.08) | 0.420(.06) | 0.472(.05) | 0.504(.05) | 0.577(.06) |
| | Ours | **0.344(.05)** | **0.576(.07)** | **0.646(.03)** | **0.686(.03)** | 0.706(.03) | 0.728(.04) | 0.778(.03) |
| SSL | PCL (Swin-Unet) [5] | 0.321(.05) | 0.556(.06) | 0.658(.03) | 0.692(.04) | 0.722(.04) | 0.747(.04) | 0.783(.03) |
| | PCL (SwinUNETR) [58] | 0.343(.06) | 0.576(.07) | 0.644(.05) | 0.692(.06) | 0.714(.05) | 0.730(.05) | 0.786(.03) |
| | Ours | **0.392(.06)** | **0.636(.06)** | **0.693(.03)** | **0.712(.03)** | **0.728(.04)** | **0.754(.04)** | **0.788(.03)** |

| Init. | Method | ACDC (100 patients in total) | | | | | | |
|---|---|---|---|---|---|---|---|---|
| | | $M$=2 | $M$=6 | $M$=10 | $M$=15 | $M$=20 | $M$=30 | $M$=80 |
| Random | SwinUNETR-3d [58] | 0.391(.07) | 0.466(.05) | 0.545(.05) | 0.603(.03) | 0.629(.03) | 0.679(.02) | 0.735(.01) |
| | Ours | **0.655(.05)** | **0.827(.05)** | **0.871(.02)** | **0.897(.01)** | **0.901(.01)** | **0.915(.00)** | **0.927(.00)** |
| SSL | PCL (Swin-Unet) [5] | 0.292(.09) | 0.577(.07) | 0.702(.04) | 0.742(.03) | 0.798(.02) | 0.825(.02) | 0.878(.01) |
| | PCL (SwinUNETR) [58] | 0.547(.05) | 0.759(.05) | 0.808(.05) | 0.850(.03) | 0.882(.01) | 0.904(.01) | 0.923(.00) |
| | Ours | **0.741(.03)** | **0.873(.01)** | **0.895(.01)** | **0.908(.01)** | **0.915(.00)** | **0.921(.00)** | **0.930(.00)** |

**Ablation on keypoint detection method** While we use SIFT as keypoint detector, it can be replaced with other learning based SoTA methods as well. We used a pretrained keypoint detection model Superpoint [13] as an alternative to SIFT. Segmentation results on CHD and ACDC dataset are reported below comparing two methods, and the results indicate that our method is not sensitive to different keypoint detection algorithms. Overall, Superpoint leads to slightly lower results, and we speculate this was due to the pretraining was done on natural images (COCO dataset).

Table 4: Segmentation results (dice scores) with different keypoint detection methods ($M$=2).

| Init. | Keypoint detector | CHD | ACDC |
|---|---|---|---|
| Random Init. | SuperPoint [13] | 0.643(.04) | 0.810(.02) |
| | SIFT | 0.686(.03) | 0.827(.05) |
| SSL Pretrain | SuperPoint [13] | 0.703(.03) | 0.865(.02) |
| | SIFT | 0.712(.03) | 0.873(.01) |

**Sensitivity of the keypoint correspondence** To verify how sensitive the method is to the number of matching keypoints, we perform an ablation on different threshold values, to pretrain our model on the CHD dataset , and finetune it on $M$=15 labeled data. Global losses ( $w_1 = w_2 = 0$) are turned off to isolate the effects of the threshold for the local SSL loss. The dice scores are reported in Table 5, indicating that our model is not sensitive to the threshold setting and remains robust across different values. We also visualize the correspondence and the number of matching keypoints under different threshold values in Fig 4.

Table 5: Dice scores under different keypoint correspondence threshold values. Overall, we observe that the results are not sensitive to specific threshold values.

| Threshold | Dice |
|-----------|------|
| 5 | 0.689 (0.043) |
| 10 | 0.690 (0.031) |
| 15 | 0.684 (0.035) |
| 20 | 0.689 (0.036) |
| 25 | 0.684 (0.032) |
| 30 | 0.685 (0.033) |
| 35 | 0.690 (0.037) |
| 40 | 0.687 (0.037) |

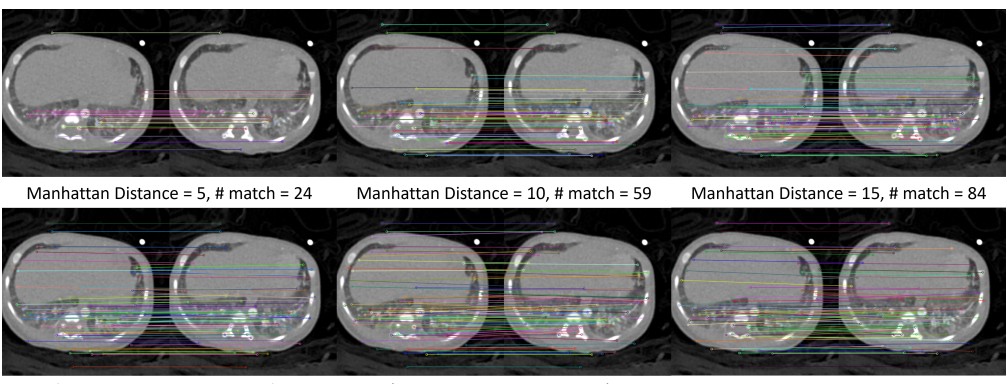

Manhattan Distance = 5, # match = 24   Manhattan Distance = 10, # match = 59   Manhattan Distance = 15, # match = 84

Manhattan Distance = 20, # match = 89   Manhattan Distance = 30, # match = 88   Manhattan Distance = 40, # match = 88

Figure 4: Sensitivity analysis of the Manhattan distance, illustrating the correspondence found under the specified Manhattan distance threshold.

**Additional dataset** We include experiments on a non-cardiac multi-organ CT segmentation dataset to evaluate the generalization of our method. Results are reported below. With 2 training subjects, we tested the performance of the model under both random and SSL pretrained initialization. In both scenarios, our method outperformed the existing works.

Table 6: Segmentation results on Synapse dataset, where finetuning is done on $M$=2 subjects out of a total of 18 subjects.

| Init. | Method | Dice (M=2) |
|-------|--------|------------|
| Random | Unet [52] | 0.253(.06) |
| | Swin-Unet [5] | 0.198(.04) |
| | SwinUNETR [58] | 0.279(.06) |
| | Ours | **0.289(.06)** |
| SSL | PCL [72] | 0.306(.05) |
| | Swin-Unet (with [72]) | 0.210(.07) |
| | SwinUNETR (with [72]) | 0.304(.06) |
| | Ours | **0.322(.06)** |

**Ablation on UNet Channels (Double Channels).** In our implementation, we augment the convolutional UNet features by concatenating them with features learned from KAF. Introducing KAF layers leads to an increase in the total number of parameters of the final KAF-enhanced UNet. Specifically, the number of channels for the second, third, and fourth blocks in the KAF-enhanced UNet become twice as large as the original UNet. For a more fair comparison, we construct a non-KAF UNet ('`UNet(c2)`' in Tab. 7) with the same amount of parameters in convolutional layers as our model by duplicating the features of the first, second, third, and fourth blocks from the UNet baseline ('`UNet(c1)`' in Tab. 7) and concatenated each with itself. The segmentation results are presented in Table 7. `UNet(c1)` indicates the UNet backbone, and `UNet(c2)` denotes the larger UNet whose convolutional parameters matched our model. They indicate that widening the network's architecture by increasing the input channel size can improve its performance. However, the performance enhancement is even more substantial with our modified layer. This suggests that incorporating features beyond simple convolution into the network architecture can further enhance the network's performance.

Table 7: Performance comparison among standard UNet (`UNet(c1)`), a larger UNet with duplicated input channels (`UNet(c2)`), and UNet augmented with features derived from KAF (Ours). The results indicate that using a larger UNet slightly improves the segmentation performance. Our proposed KAF-enhanced UNet further boosts the performance significantly compared with `UNet(c2)`.

| Sample $M$ | dataset | Method | mean/std | #params |
|---|---|---|---|---|
| 15 | CHD | UNet(c1) | 0.627(.05) | 7.8 M |
| 15 | CHD | UNet(c2) | 0.646(.04) | 27.9 M |
| 15 | CHD | Ours | 0.712(.03) | 71.7 M |
| 6 | ACDC | UNet(c1) | 0.782(.03) | 17.5 M |
| 6 | ACDC | UNet(c2) | 0.796(.03) | 62.8 M |
| 6 | ACDC | Ours | 0.873(.01) | 106.8 M |

**Ablation on Correspondence Weights.** To further investigate the contribution of the correspondence loss to the performance of the pretraining weights, we conducted a study on the various combinations of weights $w_1$, $w_2$, and $w_3$, as defined in eq. 3, and results are reported in Table 8.

First, we study the effect of varying $w_3$ by setting both $w_1$ and $w_2$ to 1. We experimented with values of 0.1, 0.01, and 0.001 for $w_3$. Among these, 0.01 yielded the best performance. In a subsequent series of experiments, we set $w_1$ to 0, intending to exclusively investigate the impact of the local correspondence loss on the global KAF self-supervised learning (SSL) loss. When only $w_2$ is set to 1, the model achieves an dice of 0.701. Upon varying $w_3$ while keeping $w_2$ fixed at 1, in most instances, incorporating $\mathcal{L}_{local}$ yields performance better than with $\mathcal{L}_{global}$ only. This implies that our local KAF correspondence SSL loss indeed offers a superior local minimum for downstream tasks. When the weights are set to 1, 1, and 0, our model's performance, at 0.689, still surpasses that of a model trained from scratch (0.686). This suggests that using only $\mathcal{L}_{local}$ could assist in finding a more optimal starting point for fine-tuning. However, when compared to having only local losses, global pretraining losses contribute more to enhancing performance.

Table 8: Additional assessment of weight of $\mathcal{L}_{local}$ in pretraining to supplement Tab. 2 in the main text. The results are all from pretraining on CHD and finetuning at $M = 15$.

| $w_1$ | $w_2$ | $w_3$ | Dice |
|---|---|---|---|
| 1 | 1 | 0.1 | 0.702(.04) |
| 1 | 1 | 0.01 | 0.712(.03) |
| 1 | 1 | 0.001 | 0.708(.04) |
| 0 | 1 | 0.08 | 0.705(.03) |
| 0 | 1 | 0.04 | 0.705(.03) |
| 0 | 1 | 0.02 | 0.707(.03) |
| 0 | 1 | 0.01 | 0.700(.03) |
| 0 | 1 | 0.005 | 0.700(.03) |
| 0 | 1 | 0.001 | 0.705(.03) |
| 0 | 0 | 1 | 0.689(.04) |

**KAF Layer in FCN.** To validate the general usefulness of the KAF layer, we injected KAF layer into a different segmentation backbone Fully Convolutional Network (FCN) [40] and compare the results between the original FCN versus the KAF-enhanced FCN. Note that in the original FCN implementation, VGG [55] was used as the encoder to gather multi-resolution features from different layers. However, given the small training size of our dataset and the relatively high complexity of VGG, we substitute the VGG encoder with a shallower CNN. Specifically, we replaced each block in VGG with two convolution layers, aiming to create a more efficient model that better suits our dataset and task.

The results of our experiment are reported in Table 9. We conduct trials with different training sample sizes. The results verify that appending KAF layers to FCN boosts performance across all sample sizes, with an average improvement exceeding 1%. Interestingly, we observed that FCN outperforms UNet in tasks involving training from scratch. However, when we incorporated the KAF layer into FCN, it did not surpass the performance of our layer applied to UNet. This could potentially be attributed to our approach in this experiment; we did not undertake hyperparameter optimization but instead directly added the layer we used in our main text to FCN. Therefore, compared to UNet, the performance improvement is relatively smaller.

Table 9: Segmentation results on CHD dataset from a random initialized FCN backbone. The column 'With KAF' indicates whether the proposed KAF layer is inserted to the backbone. The results demonstrate that the integration of the KAF layer tends to improve the mean values across different sample sizes, indicating an enhanced performance of the FCN when augmented with the KAF layer.

| Sample $M$ | With KAF | Fold 1 | Fold 2 | Fold 3 | Fold 4 | Fold 5 | Mean/Std |
|---|---|---|---|---|---|---|---|
| 2 | - | 0.2259 | 0.2133 | 0.3297 | 0.311 | 0.3516 | 0.286(.056) |
| | ✓ | 0.2517 | 0.2133 | 0.3392 | 0.3034 | 0.3794 | 0.297(.059) |
| 6 | - | 0.4441 | 0.5462 | 0.5427 | 0.5918 | 0.4854 | 0.522(.052) |
| | ✓ | 0.4495 | 0.5603 | 0.5652 | 0.6286 | 0.5535 | 0.551(.058) |
| 10 | - | 0.4866 | 0.5584 | 0.6393 | 0.6613 | 0.6094 | 0.591(.063) |
| | ✓ | 0.5632 | 0.6209 | 0.6381 | 0.6718 | 0.6383 | 0.626(.036) |
| 15 | - | 0.5938 | 0.6356 | 0.6702 | 0.6926 | 0.6499 | 0.648(.033) |
| | ✓ | 0.6157 | 0.6163 | 0.7117 | 0.6936 | 0.6537 | 0.658(.039) |
| 20 | - | 0.6318 | 0.6422 | 0.7102 | 0.7383 | 0.6356 | 0.672(.044) |
| | ✓ | 0.6382 | 0.6541 | 0.7112 | 0.7309 | 0.6806 | 0.683(.034) |
| 30 | - | 0.6339 | 0.6558 | 0.7498 | 0.7701 | 0.6685 | 0.696(.054) |
| | ✓ | 0.6841 | 0.6671 | 0.7507 | 0.7708 | 0.6973 | 0.714(.040) |
| 51 | - | 0.7125 | 0.7287 | 0.7693 | 0.7755 | 0.7559 | 0.748(.024) |
| | ✓ | 0.7087 | 0.7324 | 0.7711 | 0.7813 | 0.7661 | 0.752(.027) |

**Computation Analysis.** As transformers [60] also incorporate long-range dependencies by learning self-attention among uniformly distributed patches within the image, we compare the computational differences of a SwinTransformer [39] and our model. Fig. 5 displays a comparison of GFLOPs and GPU memory usage between our method and the SwinTransformer, given two specific variations: the edge length of the input image and the number of self-attentions within each transformer.

In the left-side plot, the GFLOPs of our method vary by a constant value (around 130 GFLOPs) as the depth of the attention map escalates. Conversely, this metric grows exponentially in the SwinTransformer. The disparity arises because our method keeps the number of keypoints constant, meaning that even as the edge length of the input image enlarges, the computation surge in the attention map remains consistent. In contrast, transformer-based methods like the SwinTransformer require a quadratic increase in computation to generate the attention map.

Our method demonstrates a slower growth rate in memory usage, particularly as the number of self-attentions increases. In the SwinTransformer, the size of the attention map correlates quadratically with the edge length of the image. However, our approach maintains a fixed number of input keypoints, which stabilizes the attention map as a constant factor in memory usage.

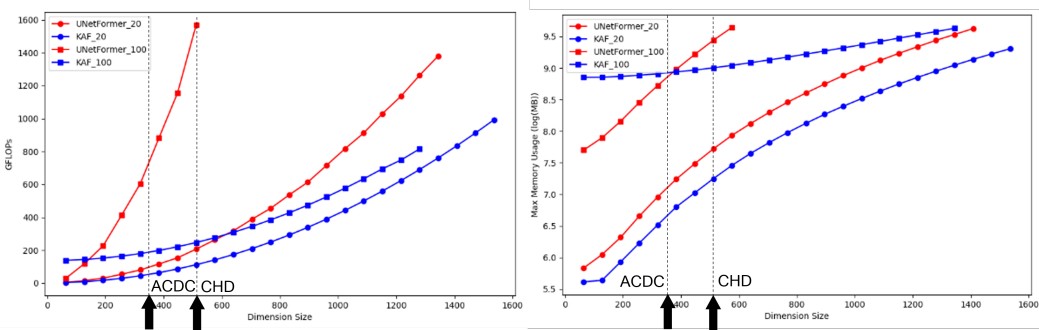

Figure 5: Comparison of GFLOPs and GPU memory usage between our method and the SwinTransformer [39]. The x-axis indicates the size of the image. We test models with different numbers of self-attention blocks (20 or 100) within the transformer, represented by different colors. The results illustrate that the GFLOPs of our method vary minimally with increasing attention map depth, and the memory usage of our method increases at a slower rate with a growing number of self-attentions. This highlights the efficiency of our method, especially with larger input image edge lengths and more complex attention maps. We also denote the actual input size of CHD dataset ($512 \times 512$) and ACDC dataset ($352 \times 352$), where our model is more computationally efficient than using a transformer in both the GFLOPs and memory consumption.

**Comprehensive 5-Fold Results.** Our detailed examination of the five folds in the CHD and ACDC datasets is exhibited in Table 10. We adhere strictly to the divisions as described in the Positional Coding Learning (PCL) study [72], ensuring consistency and reliable comparison of results.

Table 10: The complete five-fold Dice results for CHD and ACDC.

| Dataset | Sample $M$ | Fold 1 | Fold 2 | Fold 3 | Fold 4 | Fold 5 | Mean/Std |
|---------|-----------|--------|--------|--------|--------|--------|-----------|
| CHD | 2 | 0.3085 | 0.3292 | 0.4649 | 0.4405 | 0.4178 | 0.392(.062) |
| | 6 | 0.5370 | 0.6527 | 0.6707 | 0.7076 | 0.6119 | 0.636(.058) |
| | 10 | 0.6382 | 0.6797 | 0.7252 | 0.7326 | 0.6900 | 0.693(.034) |
| | 15 | 0.6668 | 0.6892 | 0.7519 | 0.7458 | 0.6844 | 0.712(.035) |
| | 20 | 0.6738 | 0.7204 | 0.7629 | 0.7766 | 0.7051 | 0.728(.038) |
| | 30 | 0.7291 | 0.7324 | 0.8001 | 0.8014 | 0.7093 | 0.754(.039) |
| | 51 | 0.7385 | 0.7594 | 0.8148 | 0.8234 | 0.8048 | 0.788(.033) |
| ACDC | 2 | 0.7975 | 0.7027 | 0.7510 | 0.7097 | 0.7458 | 0.741(.034) |
| | 6 | 0.8827 | 0.8941 | 0.8620 | 0.8596 | 0.8682 | 0.873(.013) |
| | 10 | 0.9101 | 0.9086 | 0.8919 | 0.8709 | 0.8914 | 0.895(.014) |
| | 15 | 0.9175 | 0.9076 | 0.9140 | 0.8932 | 0.9091 | 0.908(.008) |
| | 20 | 0.9173 | 0.9152 | 0.9168 | 0.9101 | 0.9143 | 0.915(.003) |
| | 30 | 0.9224 | 0.9232 | 0.9252 | 0.9162 | 0.9187 | 0.921(.003) |
| | 80 | 0.9313 | 0.9285 | 0.9336 | 0.9255 | 0.9328 | 0.930(.003) |

**Significance Tests** We include a statistical test and report p-values between results of our method versus each of the other methods in Table 11 (each scalar indicates the p-value between our final method against others). We use the average dice score per slice to estimate the p-values. All values indicate that there is a significant difference between our method and the existing methods.

Table 11: Significance tests.

| | CHD (M=15) | ACDC (M=6) |
|---|---|---|
| Unet w/ random init. [52] | < 0.001 | ≪ 0.001 |
| Ours w/ random init. | ≪ 0.001 | ≪ 0.001 |
| GLCL-full pretrain [7] | ≪ 0.001 | ≪ 0.001 |
| PCL pretrain [72] | ≪ 0.001 | ≪ 0.001 |

# B  Additional Implementation Details

**Benchmark Results** To acquire the benchmark results, we use the paper provided results whenever possible. In Table 1, results of `UNet` [52], `Rotation` [18], `PIRL` [45], `SimCLR` [9], `GLCL-global` [7], `PCL` [72] are obtained from [72], and we reproduce the remaining results including `Swin-Unet` [5], `SwinUNETR` [58], `GLCL-local` [7], `CAiD` [56] by training the model provided in their official repositories.

**Keypoint Preprocessing Details.** We compute the keypoint positions on the original image using SIFT. To obtain the keypoint positions on augmented images, we propose three potential solutions: (1) Extract the translation matrix from the augmented image and apply this translation to the keypoint positions. (2) Recompute SIFT on the augmented image to determine the new keypoint positions. (3) Consider each keypoint position as a label, and augment these labels alongside the original image.

The first solution involves retrieving the translation matrix from the augmentation package, which is not easily accessible in popular data augmentation packages currently available. Moreover, this approach requires careful handling of the cropping of the translated keypoints. The second solution poses the challenge of translating the correspondence from the original image pair to the augmented image pair. The third solution, on the other hand, retains the index of the keypoints, thus preserving the correspondence from the original to the augmented image. Consequently, we choose the third solution as our preferred approach.

To elaborate further, we initially assign a unique index to each detected keypoint. Subsequently, this index is attributed to the image space, resulting in a 2D matrix. In our implementation, the background is designated as 0, while the keypoint index starts from 1. This matrix is then transformed concurrently with the original image, resulting in a translated keypoint index matrix. We further process the keypoints under the following two circumstances: (1) If a keypoint is present in the original image gets cropped out, we simply disregard that keypoint. (2) The keypoints might project onto one or multiple nearby positions. In this scenario, we compute the mean of all these positions, and this average becomes the new keypoint position in the augmented image.

# C  Additional Results

**Keypoints Detection Results.** In Fig. 6 and 7, we present the input images and detected keypoints using the Scale-Invariant Feature Transform (SIFT) [41]. The results suggest that the majority of the detected keypoints are located in the foreground rather than the background. This helps the KAF layer to concentrate more on the regions that are crucial for segmentation tasks. Furthermore, we display the transformed keypoint positions after performing data augmentation on the input image in the third column. As described in the preprocessing details in Sec. B, during the augmentation, we transform the keypoint positions from the original image to the augmented one with the transformation matrix.

**Keypoint Correspondence on Images.** In columns (b) and (d) of Fig. 6 and Fig. 7, we provide a visual representation of the detected image correspondence. The results indicate that our heuristic distance metrics effectively identify correspondences between neighboring slices in biomedical images. During data augmentation, rather than recomputing the correspondences between the augmented slices, we translate the found correspondences from the original slices to the augmented ones, similar as the keypoint processing above.

**Additional Self-attention Map.** In Fig. 8, we provide a visualization of the self-similarity maps from two neighboring slices derived from various layers of our UNet. Similar to Fig. 3 in the main text, the similarity map of our method demonstrates better resilience to augmentations and maintains localized consistency among keypoints, compared to other pretrained models such as PCL and GLCL [7].

**Additional Segmentation Results.** To supplement Sec. 4, we present additional segmentation results comparing our method with other baselines in Fig. 9. Models trained and/or finetuned with different numbers of subjects from both ACDC and CHD datasets are present.

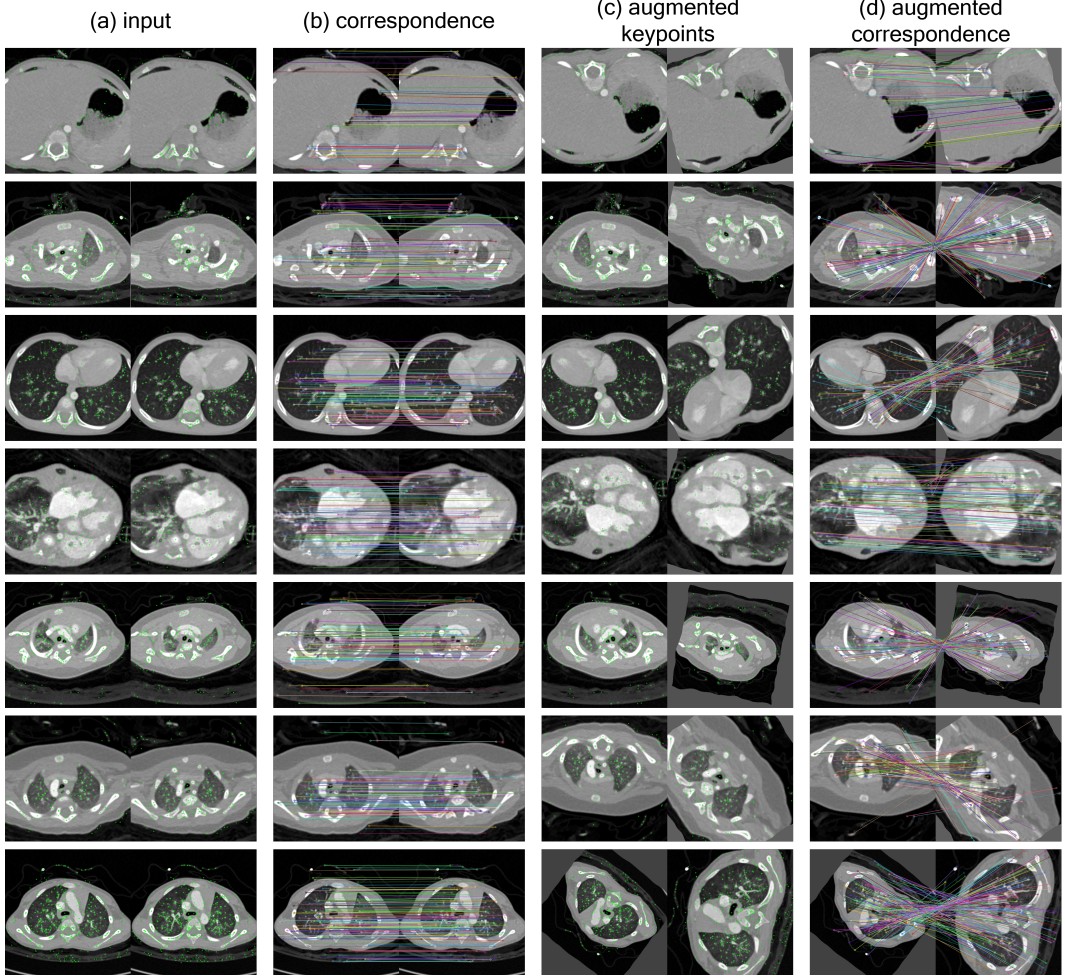

| (a) input | (b) correspondence | (c) augmented keypoints | (d) augmented correspondence |

Figure 6: Visualizatin of the detected keypoints and their correspondence on images sampled from CHD dataset. The green dots represent the keypoints detected by SIFT. In column (a), we illustrate two adjacent slices. In column (b), we showcase the correspondence between these two slices, applying the heuristic described in section 3.2 from the main text. For the actual training, we apply data augmentation into the input image, and examples are shown in (c) with the augmented keypoints. We also display the transferred correspondence in column (d). Please zoom in to view the details.

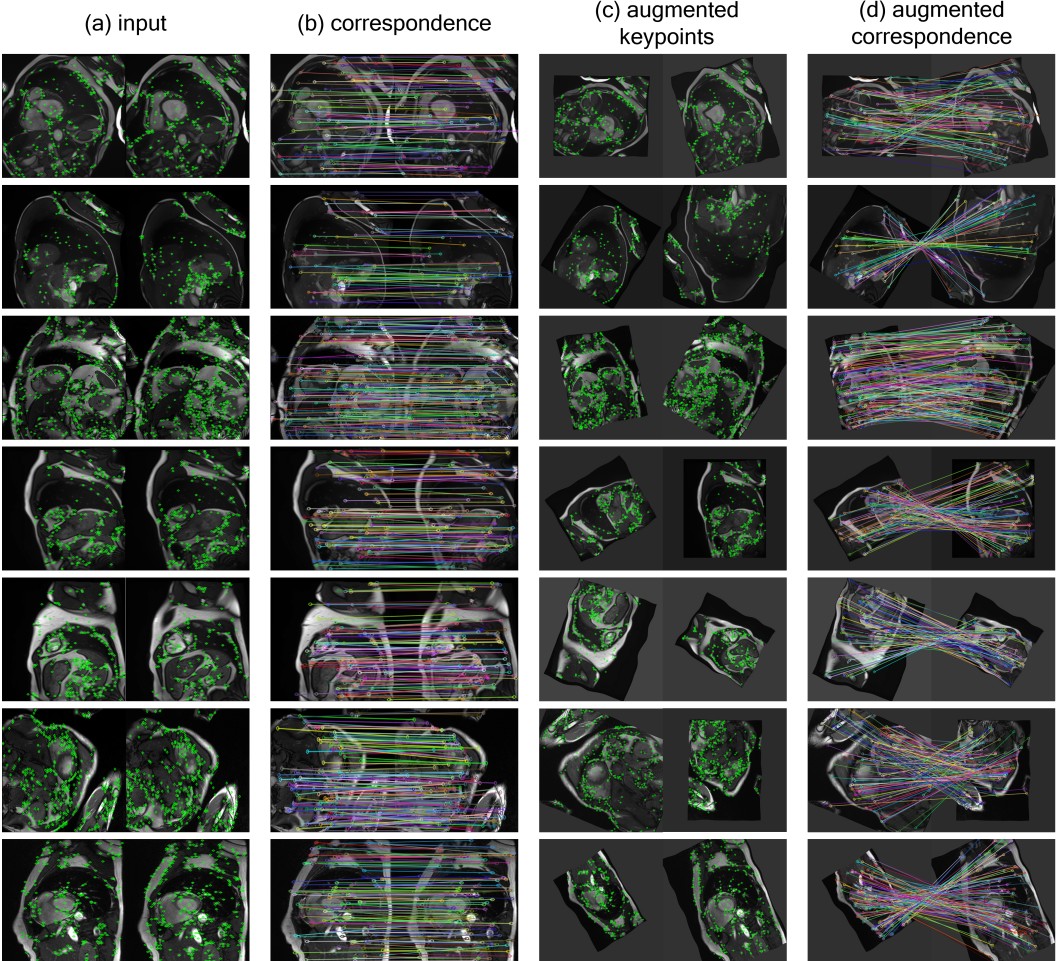

Figure 7: Visualization of slices, keypoints, and correspondences on the ACDC dataset.

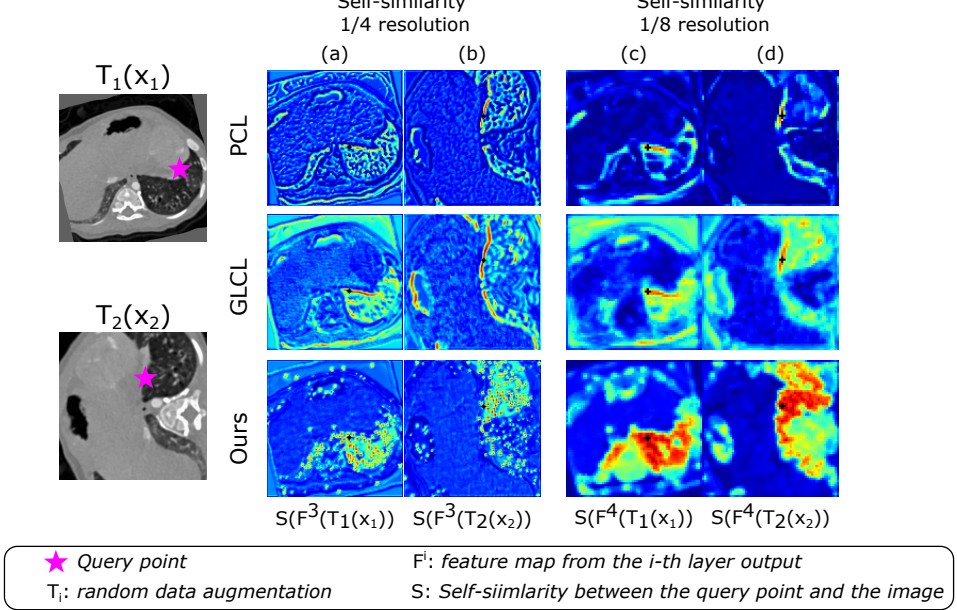

Figure 8: Learned self-similarity from two adjacent slices. Each map indicates the feature similarity between the query point feature (star) and other points within the image. We display similarity at two different scales within the UNet encoder. The comparison between (a) and (b), (c) and (d) indicates that ours is more resilient in maintaining the self-similarity of the features under various transformations.

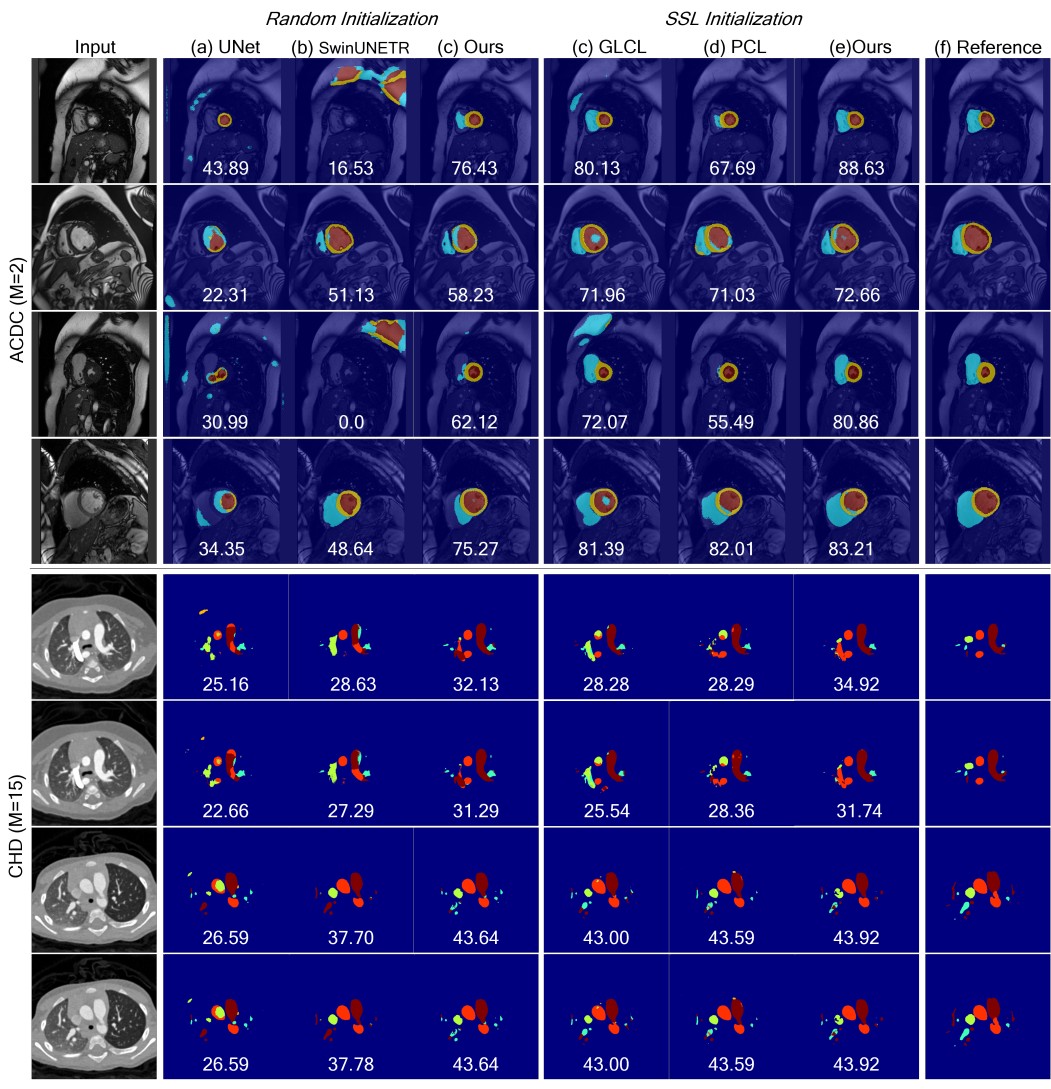

Figure 9: Additional segmentation results to supplement Fig. 2 in the main text.

