# OpenReview forum: "Keypoint-Augmented Self-Supervised Learning for Medical Image Segmentation with Limited Annotation"
_NeurIPS.cc/2023/Conference — NeurIPS 2023 poster_

### Official Review · Reviewer_PaeV · 2023-06-23

**Soundness:** 3 good
**Presentation:** 3 good
**Contribution:** 2 fair
**Rating:** 6
**Confidence:** 4

**Summary:**

This paper tackles self-supervised learning for medical image segmentation. The main contribution is the keypoint identification in input images in self-supervised pre-training to consider long-range spatial dependencies. Each UNet encoder layer is followed by a keypoint-augmented fusion (KAF) layer, where keypoints are identified by SIFT on the input image and rescaled with respect to the resolution of each layer. Convolutional features at each keypoint are processed by a Vision Transformer to incorporate self-attention, provided to additional convolutional layers for extracting dense features from transformer outputs and finally concatenated with the input feature map. Resulting KAF layer features are used for contrastive training of UNet for both global representation learning similar to PCL [54], as well as local similarity learning via the SuperGlue graph neural network [42].

**Strengths:**

Incorporating similarity learning from keypoints is interesting and novel in self-supervised medical image segmentation literature. Paper is well-written and the experiment setup is detailed. Cross-validation is used to compute error bounds in addition to performance metrics. Downstream segmentation performance with limited annotations outperforms several pre-training approaches across two benchmark datasets.

**Weaknesses:**

While contributions are novel, literature review is missing a lot of works on self-supervised medical image segmentation via global/local similarity/contrastive learning:

Taher et al. "CAiD: Context-Aware Instance Discrimination for Self-supervised Learning in Medical Imaging" 2022

Yan et al. "SAM: Self-supervised Learning of Pixel-wise Anatomical Embeddings in Radiological Images" 2021

Zheng et al. "MsVRL: Self-Supervised Multiscale Visual Representation Learning via Cross-Level Consistency for Medical Image Segmentation" 2023

Ouyang et al. "Self-supervised Learning for Few-shot Medical Image Segmentation" 2022

Fischer et al. "Self-supervised contrastive learning with random walks for medical image segmentation with limited annotations" 2023

Xie et al. "PGL: Prior-Guided Local Self-supervised Learning for 3D Medical Image Segmentation" 2020

Wang et al. "Self-supervised learning based transformer and convolution hybrid network for one-shot organ segmentation" 2022: also incorporates Vision Transformer for global feature learning.

Experimental comparisons against Wang et al. 2022 would further strengthen the novelty.


**Questions:**

How are the thresholds selected for identifying positive/negative keypoints?


**Limitations:**

limitations and future work are discussed.

---

> ### Author Rebuttal · Authors · 2023-08-10
>
>
> > While contributions are novel, literature review is missing a lot of works on self-supervised medical image segmentation via global/local similarity/contrastive learning.
> > Experimental comparisons against Wang et al. 2022 would further strengthen the novelty.
>
>
> Thank you for offering additional literature on self-supervised medical image segmentation! We will include them in our related works. Unfortunately, we were not able to find the publicly available source code for `Wang et al.`, but we are happy to include them as one of the comparisons when the authors release the code in the near future.
>
>
> Among the list of papers provided, we were only able to find one available (`Taher et al.`) codebase, and we include them as an additional comparison.
> Results on CHD and ACDC dataset are reported below. Our method outperforms CAiD in both datasets under different numbers of subjects used for finetuning (M). A full comparison will be further attached into the revised paper.
>
>
> | | CHD | | ACDC | |
> |------------|:---:|------------|:----:|------------|
> | SSL Method | M=2 | M=15 | M=2 | M=6 |
> | *CAiD* | 0.265(.08) | 0.684(.04) | 0.483(.11) | 0.822(.02) |
> | Ours | **0.392(.06)** | **0.712(.03**) | **0.741(.03)** | **0.873(.01)** |
>
>
>
>
>
>
> > How are the thresholds selected for identifying positive/negative keypoints?
>
>
> As there are two thresholds involved in our models, we clarify each of them below.
> - The first threshold is applied across different slices to determine the positive/negative slices, and we follow the optimal value found by the PCL paper [1] - 0.1 and 0.35 for CHD and ACDC.
> - The second threshold is applied among keypoint features between two positional positive slices to determine the correspondence among these points. In this step, we do not explicitly recognize any "negative" keypoints, as we only use the matched correspondence ("positive") in our local loss. By default, we set the threshold as 20. From our ablation on the threshold, and the attached visualization of the correspondence, our method is not sensitive to the threshold values and the number of the matched correspondence.
>
>
> [1] Zeng et al., Positional contrastive learning for volumetric medical image segmentation.

---

> > ### Comment · Reviewer_PaeV · 2023-08-14
> >
> > I have read the rebuttal and keep my original score.
> >
> > Thank you,

---

> > > ### Author Response · Authors · 2023-08-18
> > >
> > > Dear Reviewer PaeV,
> > >
> > > Thanks a lot for the positive feedback!

---

### Official Review · Reviewer_8g4p · 2023-07-04

**Soundness:** 3 good
**Presentation:** 3 good
**Contribution:** 3 good
**Rating:** 5
**Confidence:** 3

**Summary:**

This paper focuses on improving medical image segmentation performance with limited labeled data. They (1) propose a modification to the standard UNet convolutional architecture by incorporating local features (at locations identified by SIFT) into the backbone, (2) show how their modification can be used to control self-supervised contrastive learning positive and negative pair selection, and (3) evaluate their proposed architectural change and self-supervised training strategy against relevant baselines. They find both the architectural change and self-supervised strategy improve segmentation performance.

**Strengths:**

- Novel architectural change that impacts network performance both with random initializations and with self-supervised contrastive learning.
- Improves performance against relevant and recent self-supervised baselines.


**Weaknesses:**

- The GLCL baseline reported in Table 1 is much lower than the numbers reported in the GLCL original paper; the original GLCL paper actually reports performance better than the proposed method on the ACDC dataset. Please comment on why the reported GLCL performance in 14-18 points lower than the GLCL paper.
- The proposed architectural change may increase compute requirements, but this is not discussed nor are any timing/computational load experiments reported.
- There is limited dataset variety—both evaluation datasets are 3D cardiac anatomical segmentation tasks. Additional experiments, particularly including pathological segmentation, would strengthen the results and show the proposed method works for different segmentation targets.
- Scaling the proposed solution to non-CNN architectures or 3d data may be difficult (as mentioned by the authors).
- Minor: Table 1 caption needs to specify what metric you are reporting; Line 221 has a period where there should be a comma+”and.”



**Questions:**

My questions and suggestions are covered in the Weaknesses section.

**Limitations:**

Limitations and impact adequately discussed.

---

> ### Author Rebuttal · Authors · 2023-08-10
>
> > The GLCL baseline reported in Table 1 is much lower than the numbers reported in the GLCL original paper; the original GLCL paper actually reports performance better than the proposed method on the ACDC dataset. Please comment on why the reported GLCL performance in 14-18 points lower than the GLCL paper.
>
>
>
>
> Thank you for pointing this out! We believe this is a miscommunication on our part and we will clarify in our paper. In the original GLCL paper, experiments are conducted based on a single fold train/test split, and results are reported on a single held-out test set. In our experiment, we followed the PCL paper and conducted 5-fold cross validation. Results in Table 1 indicate an average dice score across the 5 folds. Note that our reported result of the GLCL-global is consistent with the GCL results reported in the PCL paper (where they adopted the global loss of the GLCL). We additionally performed 5 folds evaluation with the full losses and results are shown in Table 1.
>
>
>
>
>
>
>
>
> > The proposed architectural change may increase compute requirements, but this is not discussed nor are any timing/computational load experiments reported.
>
>
>
>
> Thank you for your suggestion! Due to the page limit, we discussed the computational load in the supplementary materials. Please kindly refer to appendix A "Computational analysis" for more information.
> In summary, our method demonstrates a slower growth rate in both memory usage and GFLOPs compared with transformer-based unet (i.e. SwinUNETR).
>
>
>
>
> > Limited dataset variety.
>
>
> We agree. We now include results on an additional CT dataset for multi-organ segmentation. Results are reported in the Table below. With 2 training subjects, we tested the performance of the model under both random weight initialization and SSL pretrained initialization. In both scenarios, our method consistently outperformed the existing works. Due to the time constraint, we will supplement the remaining methods and different number of training subjects in the revised manuscript.
>
>
> | Init. | Method | Dice (M=2) |
> |--------------|----------------------|------------|
> | Random Init. | Unet | 0.253(.06) |
> | | Swin-Unet | 0.198(.04) |
> | | SwinUNETR | 0.279(.06) |
> | | Ours | 0.289(.06) |
> | SSL pretrain | PCL | 0.306(.05) |
> | | Swin-Unet (with PCL) | 0.210(.07) |
> | | Ours | 0.322(.06) |
>
>
>
>
>
>
> > Scaling the proposed solution to non-CNN architectures or 3d data may be difficult (as mentioned by the authors).
>
>
>
> Thank you for the comment. It is worth mentioning that the current scope of our work is to enable the long-range dependencies into the CNN based segmentation architectures, and our results on the 2D medical image segmentation indicates the effectiveness of the proposed keypoint-based formulation. Our proposed KAF layer is generic, which can also be incorporated into recent transformer-based Unet architectures that integrate both transformer blocks and the convolutional blocks. We fully agree that such architectural modification or scaling up to 3D may require different consideration and further investigation, but these directions are interesting future works.
>
>
>
>
> > Table 1 and Line 221
>
>
>
>
> Thanks for pointing that out, and we will mention that we use Dice as a metric and report its computed mean and standard deviation in Table 1.
> For Line 221, we will change the dot to period. Thanks!

---

> > ### Comment · Reviewer_8g4p · 2023-08-11
> > **Rebuttal response**
> >
> > I have read the other reviews and the authors' rebuttal. The authors have presented a thorough rebuttal responding to the raised questions/weaknesses and have presented new data showing useful extensions and ablations of their method. I think this is an interesting addition to the self-supervised body of work and am changing my score from 4 to 5.

---

> > > ### Author Response · Authors · 2023-08-18
> > >
> > > Dear Reviewer 8g4p,
> > >
> > > Thank you for your time and we appreciate your encouragement! Please let us know if you have other comments or questions.

---

### Official Review · Reviewer_m84B · 2023-07-05

**Soundness:** 3 good
**Presentation:** 3 good
**Contribution:** 3 good
**Rating:** 6
**Confidence:** 5

**Summary:**

SIFT keypoints are used in conjunction with a UNET to construct a transformer-like model for medical image segmentation. Global and local self supervised learning (SSL) losses are proposed. Correspondences identified using the SuperGlue method. Various experiments on two cardiac image datasets, including ablation studies, shows improves on other work in terms of segmentation dice score.



**Strengths:**

The use of traditional keypoints as attention operator mechanism for UNet segmentation makes sense and is novel, to my knowledge.

Improved segmentation results convince the reader that the approach has practical value.

**Weaknesses:**

The method presented is based on 2D keypoint matching, issues with multiple 2D slices, wheras anatomy is 3D. It would be interesting to know if 3D keypoints would improve results.

Table 1 should mention that the numbers shown are dice scores.

Minor spelling
Line 182 “based on the cloest L2 distance” -> “closest”
Line 213 “patent ages ranging from 1 month to 21 year” -> “21 years”

It would be interesting to know much the learning method here improves upon simple keypoint transfer segmentation as in [46].

Objective functions in equations (1) and (2) appear very similar to other recent keypoint matching work:
Chauvin, Laurent, et al. "Efficient pairwise neuroimage analysis using the soft jaccard index and 3d keypoint sets." IEEE transactions on medical imaging 41.4 (2021): 836-845.

The description of the datasets used does not mention the size and resolution of the images.

Computational complexity of the proposed method is not mentioned.

The paper lacks reproducibility. For instance, line 156 and line 181 allude to thresholds whose numerical values are not explicitly mentioned. Similarly, it is not clear what setup was used to train the Transformer block, including details such as learning rate, batch size, attention head size, and input embedding size.

Additionally, it would be valuable to include statistical tests to demonstrate the superiority of the proposed method compared to existing approaches.


**Questions:**

- It would be good to hear a discussion regarding 3D keypoint methods, and how much this method would improve upon simple keypoint transfer segmentation as in [46].

- In lines 208 and 209 it is stated that images are pre-processed using the methodology described in [54]. Could the preprocessing stage be briefly described in the paper? Are there aspects without which this approach would not work, e.g. registration of training data?
- Could the authors explain why the Manhattan distance is preferred over other distance measures for keypoint filtering (as mentioned in line 181)?
- Would it be possible for the authors to provide the values of the similarity index in Figure 2? This would allow us to compare those values with the ones presented in the previous Table 1.
- Can the authors provide more insight into the justification for the following claim? Clearly there are various random transforms which would scramble image information, to which SIFT is not invariant.
"(2) our proposed self-supervision helps maintain better local equivariance of the self-attention, i.e., with the  same query point location, its interaction with other points remains identical no matter how the image is transformed."


**Limitations:**

Several limitations exist: missing important details, computational complexity, description of data, transformer parameters (learning rate, batch size, attention head size, and input embedding size), statistical significance and comparison to based keypoint transfer segmentation.

---

> ### Author Rebuttal · Authors · 2023-08-10
>
> > It would be interesting to know if 3D keypoints would improve results.
>
> Thank you for the insightful point! As discussed in our limitations section, a promising future avenue is expanding our model to 3D. This direction, however, poses unique challenges. Firstly, 3D keypoint detection lacks established algorithms. Moreover, the inherent sparsity of 3D data adds complexity, making the transfer of features from sparse keypoint maps to dense feature maps a formidable task. Modifying this module presents an unexplored area requiring careful investigation.
> Furthermore, the increase in keypoints from $O(x^2)$ to $O(x^3)$ brings about the necessity for a novel sampling strategy to address memory constraints. While deploying our method on 3D data holds potential, we acknowledge the substantial learning curve and the need for extensive experimentation in this domain.
>
> > It would be interesting to know how much the learning method here improves upon simple keypoint transfer segmentation as in [46].
>
> We fully agree that computing a registration based on keypoints and then transforming the segmentation task to the target image is an interesting idea when annotations are limited. However, we note that extending [46] for our task requires extensive research efforts that leads to a different project. Crucially, [46] registers key points across different subjects in order to transfer training segmentation during test time, whereas in our method, keypoint correspondence is detected across neighboring slices within the same subject, which is much simpler and trivial. Moreover, [46] uses keypoints for segmentation transfer, while our method utilizes keypoints for more general-purpose representation learning. Again, we acknowledge that comparing or even extending [46] under our task will be a promising future research direction.
>
> > Objective functions in equations (1) and (2) appear very similar to other recent keypoint matching work.
>
> Yes, we agree. In our approach, eq (1) is an established global contrastive loss, and the distinction of our method with others lies in the features used to compute similarity. Specifically, the global feature is now computed from our KAF layer in eq(1). Similarly, we follow SuperGlue in eq(2) for aligning the feature space between the matched correspondence features.
>
> [1] Sarlin et al.,  SuperGlue: Learning Feature Matching With Graph Neural Networks
>
> > The size and resolution of the images.
>
> We mentioned the resolution of the image size in the caption of Figure 1 in our appendix, but we will make it more explicit in the main text.
>
> > Computational complexity.
>
> Please kindly refer to our appendix A "Computational analysis".
>
> > Reproducibility: Threshold values in L156 and L181, learning rate, batch size, etc.
>
> Thank you for raising the issues. We included learning rate, batch size, optimizers in L226 – L240, but we are happy to add more relevant details in the manuscript. Our code will be released for better reproducibility.
>
> > Statistical tests.
>
> Thanks for the valuable suggestion. We include a statistical test and report p-values between results of our method versus each of the other methods below (each scalar indicates the p-value between our final method against others). We use the average dice score per slice to estimate the p-values. All values indicate that there is a significant difference between our method and the existing methods.
>
> | | CHD (M=15) | ACDC (M=6) |
> |----------------------|------------|------------|
> | Unet w/ random init. | < 0.001 | << 0.001 |
> | Ours w/ random init. | << 0.001 | << 0.001 |
> | GLCL-full pretrain | << 0.001 | << 0.001 |
> | PCL pretrain | << 0.001 | << 0.001 |
>
> > In lines 208 and 209 it is stated that images are pre-processed using the methodology described in [54]. Could the preprocessing stage be briefly described in the paper? Are there aspects without which this approach would not work, e.g. registration of training data?
>
> We will add a description on the preprocessing. It mainly included intensity normalization and resampling, which is a commonly adopted preprocessing step in medical images. Further, our model does not assume registered training images and thus won’t be affected by such pre-processing.
>
> > Why is the Manhattan distance preferred over other measures?
>
> We empirically used Manhattan distance as the criteria inspired by the PCL paper, but it is likely that other measures can also work, as long as the correspondence can be reasonably determined. This assumption was also supported by our ablation study for `9qHo` using different thresholds, where the number of matched correspondences and different threshold values had minimum effects on the final results.
>
> > Adding values of the similarity index in Figure 2.
>
> We now added the corresponding Dice (scaled by 100)  into the revised figure attached. We observed that these individual values are consistent with the average scores in Table 1.
>
> > Insights on (1) SIFT is not invariant. (2) “our proposed self-supervision helps maintain better local equivariance of the self-attention."
>
> We fully agree that SIFT may not be robust when strong image augmentations are applied on the images, and this is precisely the reason why we first detect keypoints on the **original** images before any augmentations are applied, and this makes sure that the keypoints are consistent under random augmentations. Further, the keypoint correspondences are also matched on the original images, and our local loss further encourages that the features between corresponding points remain similar even under strong augmentations. Such formulation ensures that our model is more robust in maintaining the identical interaction between keypoints under different image transformations, as shown in main text Figure 3 and appendix Figure 2.

---

> > ### Comment · Reviewer_m84B · 2023-08-14
> >
> > I was initially very enthusiastic about this work, I am generally satisfied with the authors responses. Unfortunately the authors miss directly related literature and avoid broadening their paper literature review suggested by several reviewers, which ultimately reduces the potential impact of this work. For example, traditional 3D keypoint segmentation methods, published in a major medical imaging journal.
> >   Wachinger, C. et al. (2018). Keypoint transfer for fast whole-body segmentation. IEEE transactions on medical imaging, 39(2), 273-282.
> >
> > I thus reduce my recommendation from 7 to 6, more inline with other reviewers.

---

> > > ### Author Response · Authors · 2023-08-18
> > >
> > > Dear Reviewer m84B,
> > >
> > > Thank you for your time and we appreciate your overall positive and constructive feedback after reading our rebuttal!
> > >
> > > We believe there might be some miscommunication on our part and we would like to take this opportunity to provide further clarity regarding the related literature and the potential impact of our research.
> > >
> > > > “The authors miss directly related literature and avoid broadening their paper literature review suggested by several reviewers that ultimately reduced the impact of this work.”
> > >
> > > We greatly respect and value the important related literature provided by all reviewers w.r.t. Transformer-based UNet (Reviewer `9qHo`), local SSL (Reviewer `b18J,PaeV`), and keypoint-based segmentation (Reviewer `m84B`). Our intention was never to overlook the importance of broadening our literature review, and we genuinely value your suggestions. We will incorporate the missing literature into our manuscript.
> > >
> > > Additionally, we have carried out extra experiments with the suggested papers whose code was accessible, to further validate our methods. To specify, beyond our initial experimental set up, we have added more experiments and comparisons with these literature as follows:
> > >
> > >  (1) Apart from the original computational comparison with transformers, we have further compared the segmentation performance with two transformer-based methods. Our architecture achieved better performance under both random initialized weights and self-supervised pretraining, while maintaining lower computational costs. (see response to `9qHo`);
> > >
> > > (2) We have added more comparisons with two recent local self/semi-supervised learning methods. Our method outperformed the existing works, which validated the benefits of our proposed pretraining strategies (see response to `b18J,PaeV`).
> > >
> > > (3) We have included a new CT dataset and verified that our method also generalizes to multi-organ non-MRI datasts (see response to `9qHo,8g4p,b18J`).
> > >
> > > We respectfully believe that these new experiments and comparisons could potentially contribute to a broader impact of our work.
> > >
> > >
> > >
> > >
> > > > "Miss directly related work on traditional 3D keypoint segmentation methods, published in a major medical imaging journal. Wachinger, C. et al. (2018). Keypoint transfer for fast whole-body segmentation. IEEE transactions on medical imaging, 39(2), 273-282"
> > >
> > > We genuinely acknowledge the significance of the work that employs keypoints for 3D segmentation, as highlighted in your comment. We did not intend to exclude a direct comparison with it, whereas a proper re-implementation without their official codebase may require extensive efforts. We are more than willing to reach out to the authors to explore the possibility of re-implementing their method for a fair comparison with ours. Moreover, we believe the 3D keypoints used in this paper will serve as a promising starting point for extending our model from 2D to 3D.
> > >
> > > Once again, we thank you for your time and effort in providing feedback. We hope that our clarification effectively addresses your concerns related to the missing literature and the potential impact of our work.

---

### Official Review · Reviewer_9qHo · 2023-07-06

**Soundness:** 2 fair
**Presentation:** 2 fair
**Contribution:** 2 fair
**Rating:** 3
**Confidence:** 4

**Summary:**

The authors introduce an approach to improve the performance of the UNet model, specifically in the task of segmentation. They propose a keypoint augmented fusion layer to enhance the CNN-based encoder of UNet. This layer leverages self-attention to capture long-range spatial dependencies among keypoints extracted from multi-scale feature maps. In addition, the paper suggests the use of global and local contrastive losses to pretrain the model in a self-supervised manner. These losses help the model learn meaningful representations from unlabeled data, improving its performance. The proposed framework was evaluated on Cardiac CT and MRI datasets for segmentation tasks. The results demonstrate reasonable improvements over the state-of-the-art methods, particularly in scenarios with limited labeled data (few-shot settings).

**Strengths:**

1) This work addresses limitations of traditional UNet models, which rely on convolutional neural networks (CNNs) and may struggle to capture long-range dependencies among spatial positions in an image. In this study, the authors propose a more effective approach to handle this issue. At each scale of the encoder, the authors employ Scale-Invariant Feature Transform (SIFT) to detect key points within the CNN feature maps. By applying SIFT, the authors aim to identify salient locations within the feature maps that can capture important spatial information. To facilitate correlation and interaction among these detected key points, a transformer model is employed.

2) Global contrastive loss is applied in an unique way on two types of representations - concatenated keypoint enhanced feature maps, as well as features  from last layer of UNet.

3) instead of being used as a traditional image-wise similarity loss, here the local contrastive loss is proposed to exploit pixel-wise similarity (by measuring the correlation between keypoints within a spatial distance among adjacent slices)

**Weaknesses:**

1) It is unclear why this work ignores Transformer-based UNet architectures (TransBTS [1], UNetr[2], Swin-Unet [3], etc) which employ self-attention to better capture the correlations. Though the authors propose to use it on keypoints, I find it difficult to comprehend the advantages here, except maybe computational efficiency. There are many works which do it on entire feature maps - why are those suboptimal?

2) Handcrafted features like SIFT/SURF to detect keypoints, though powerful, aren't the current SOTA. The performance can definitely be tested using finetuned object detection models (such as Faster RCNN [4])

In short - there should be ablation showing (1) advantage of using keypoints, (2) Different keypoint
detection strategies, and most importantly (3) comparison with more recent Transformer-based UNet architectures.

[1] TransBTS: Multimodal Brain Tumor Segmentation Using Transformer
[2] UNETR: Transformers for 3D Medical Image Segmentation
[3] Swin-Unet: Unet-like Pure Transformer for Medical Image Segmentation
[4] Faster R-CNN: Towards Real-Time Object Detection with Region Proposal Networks

**Questions:**

1) Keypoint Robustness - The manner of application of global contrastive loss depends on the effectiveness of keypoints. Because the other global loss is used off-the-shelf from [54] as PCL loss.

2) As for local loss, the contribution appears to be more engineering (lines 175-200) i.e on how to use it among keypoints in adjacent slices. Here also, there is no ablation on the multiple thresholds selected in the form of spatial distance between keypoints or number of adjacent slices.

3) Standard non-cardiac benchmark datasets like BTCV, SegTHOR have not been used. To establish it as a generic method, the authors
should definitely explore non-cardiac pan-organ datasets

4) I also fail to understand what customization the authors have proposed in their architecture to make it work in few-shot settings with limited annotations.

5) Can the authors please mention the unlabeled datasets used for SSL pretraining?

**Limitations:**

Not adequately addressed in the 'Discussion' section

---

> ### Author Rebuttal · Authors · 2023-08-10
>
> > Comparison with transformer-based UNets.
>
> Thank you for your suggestions!
> We now added two recent SoTA transformer-based unets including Swin-Unet[1] and Swin-UNETR[2]. To illustrate the effects of architectural changes, we train all models with random weight initialization, under limited annotation (number of training subjects $M=2$). Five-fold cross validation is performed on different datasets, and the average dice score with standard deviation is reported below. When trained with limited labels, Unet still outperformed transformer based architecture (except for Swin-Unet on CHD), and our method achieved the best results across the board. We note that our observation is consistent with the conclusion in [1], where transformers may be more severely affected by the initialization, and may benefit from large-scale pretraining. E.g., on Synapse, we observed an increased dice of SwinUNETR from 0.198(.04) to 0.210(.07) when further pretraining it with PCL.
>
> | | CHD | ACDC | Synapse |
> |--------------|------------|------------|------------|
> | Swin-Unet[1] | 0.236(.09) | 0.327(.10) | 0.198(.04) |
> | SwinUNETR[2] | 0.137(.09) | 0.501(.03) | 0.279(.06) |
> | Unet | 0.184(.06) | 0.588(.07) | 0.253(.06) |
> | Ours | 0.344(.05) | 0.655(.05) | 0.289(.06) |
>
>
> [1] Cao et al., "Swin-Unet: Unet-like Pure Transformer for Medical Image Segmentation".
> [2] Hatamizadeh et al., "Swin UNETR: Swin Transformers for Semantic Segmentation of Brain Tumors in MRI Images".
>
>
>
>
> > The advantage of using keypoints? The keypoint robustness? Other keypoint detection methods?
>
> Thank you for the questions. We will try to clarify our intuition with additional experiments, to hopefully address the concerns.
>
>
> **Why use keypoints?**
>
> Our rationale for using keypoints is rooted in enhancing the conventional CNN layers with long-range dependencies, a characteristic often encountered in medical images. While the attention in transformers facilitates the learning of such long-range dependencies, it operates among all pairs of feature points, potentially resulting in both computational expense and a deficiency in targeting crucial regions. Empirically, our formulation leads to much better segmentation performance compared with CNN-only UNet and transformer-based UNet, while being less computational intensive.
>
>
> **Other keypoint detection methods?**
> We agree. The keypoint detection can be replaced with other learning based SoTA methods as well. Given that finetuning Faster RCNN model for keypoint detection requires groundtruth labels (we do not have access to), we used a pretrained keypoint detection model Superpoint [1] as an alternative to SIFT. Segmentation results on CHD and ACDC dataset are reported below comparing two methods, and the results indicate that our method is not sensitive to different keypoint detection algorithms. Overall, Superpoint leads to slightly lower results, and we speculate this was due to the pretraining was done on natural images (COCO dataset).
>
>
> | Init. | | CHD | ACDC |
> |----------------|---------------------|--------------|-------------|
> | Random Init. | SuperPoint | 0.643(.04) |0.810(.02) |
> | | SIFT | 0.686(.03) |0.827(.05) |
> | SSL Pretrain | SuperPoint | 0.703(.03) |0.865(.02) |
> | | SIFT Pretrain | 0.712(.03) |0.873(.01) |
>
>
> [1] DeTone et al., "SuperPoint: Self-Supervised Interest Point Detection and Description".
>
>
>
>
> **Keypoint Robustness - No ablation on the thresholds.**
> To verify how sensitive the method is to the number of matching keypoints, we perform an ablation on different threshold values, to pretrain our model on the CHD dataset, and finetune it on $M=15$ labeled data. Global losses ($w_1=w_2=0$) are turned off to isolate the effects of the threshold for the local SSL loss. The dice scores are reported below, indicating that our model is not sensitive to the threshold setting and remains robust across different values. We also include a visualization in the attached pdf with the number of matching keypoints annotated under different threshold values.
>
>
> | Threshold | Dice |
> |:---------:|:-------------:|
> | 5 | 0.689 (0.043) |
> | 10 | 0.690 (0.031) |
> | 15 | 0.684 (0.035) |
> | 20 | 0.689 (0.036) |
> | 25 | 0.684 (0.032) |
> | 30 | 0.685 (0.033) |
> | 35 | 0.690 (0.037) |
> | 40 | 0.687 (0.037) |
>
>
>
>
> > Non-cardiac pan-organ datasets
>
>
> We now add a multi-organ CT segmentation dataset to evaluate the generalization of our method. Results are reported below. With 2 training subjects, we tested the performance of the model under both random and SSL pretrained initialization. In both scenarios, our method outperformed the existing works. We will supplement the remaining methods and different number of training subjects in the revised manuscript.
>
>
> | Init. | Method | Dice (M=2) |
> |--------------|----------------------|------------|
> | Random Init. | Unet | 0.253(.06) |
> | | Swin-Unet | 0.198(.04) |
> | | SwinUNETR |0.279(.06) |
> | | Ours | 0.289(.06) |
> | SSL pretrain | PCL | 0.306(.05) |
> | | Swin-Unet (with PCL) | 0.210(.07) |
> | | Ours | 0.322(.06) |
>
>
>
>
> > What customization in the architecture makes it work in few-shot settings?
>
> Our architectural customization involves integrating the KAF layer into every CNN block of the UNet. Leveraging the interplay among the features extracted from identified keypoints, this modification empowers the model to more effectively gather local information. Consequently, it facilitates enhanced representation learning even when working with limited data. These advantages have been substantiated through comprehensive empirical experiments conducted across diverse datasets.
>
>
> > The unlabeled datasets used for pretraining?
>
>
> We follow common practice of medical SSL and use all unlabeled volumes on each dataset for our pretraining. On ACDC, we use all unlabeled volumes (100 patients, and each of them has ~15 volumes)l. On CHD, we use a total of 68 cardiac images.

---

> > ### Author Response · Authors · 2023-08-18
> >
> > Dear Reviewer 9qHo,
> >
> > We would like to thank you again for your constructive comments and suggestions. We've expanded upon the last table above by including a more thorough comparison between our approach and the two transformer UNet models under both random and self-supervised initialization. We hope that the updated results will provide more insights w.r.t. the comparisons with transformers.
> >
> > In particular, we further pretrain the SwinUNETR with two SSL strategies: PCL [1] and the self-supervised loss proposed in [2]. Our findings are in alignment with the trends we observed earlier, wherein the performance of transformer models is notably influenced by their initialization. Both of these pretraining methods led to observable improvements, and our proposed method achieved the best performance.
> >
> >
> > | Init. | Method | Dice (M=2) |
> > |--------------|----------------------|------------|
> > | Random Init. | Unet | 0.253(.06) |
> > | | Swin-Unet | 0.198(.04) |
> > | | SwinUNETR | 0.279(.06) |
> > | | Ours | 0.289(.06) |
> > | SSL pretrain | PCL | 0.306(.05) |
> > | | Swin-Unet (with PCL) | 0.210(.07) |
> > | | SwinUNETR (pretraining with [1]) | 0.284(.07) |
> > | | SwinUNETR (PCL) | 0.304(.06) |
> > | | Ours | 0.322(.06) |
> >
> > [1] Zeng et al., “Positional Contrastive Learning for Volumetric Medical Image Segmentation”
> > [2] Tang et al., “Self-Supervised Pre-Training of Swin Transformers for 3D Medical Image Analysis”

---

> > > ### Comment · Area_Chair_honq · 2023-08-18
> > > **Rebuttal to 9qHo**
> > >
> > > Dear 9qHo
> > >
> > > Could you have a look at the rebuttal to see if your questions have been clarified?
> > >
> > > Thanks,
> > > Your AC

---

> > > ### Comment · Reviewer_9qHo · 2023-08-20
> > >
> > > Thank you for the responses. Based on the rebuttal, I have updated my score.

---

### Official Review · Reviewer_b18J · 2023-07-07

**Soundness:** 4 excellent
**Presentation:** 4 excellent
**Contribution:** 3 good
**Rating:** 6
**Confidence:** 4

**Summary:**

The paper presents a self-supervised learning approach for medical image segmentation that uses matching keypoints to contrast local features that may be spatially distant from one another. The proposed model pretrains a segmentation network on unlabelled images, using three contrastive losses: a global loss on the keypoint-enhanced features, a global loss on pooled features in the last layer (as in PCL) and a local loss on features of matching keypoints (which are spatially close and have a high matching probability). This model exploits Keypoint-Augmented Fusion (KAF) layers which transforms the features of detected keypoints using self-attention transformer layers and then concatenates the transformed features to the original feature map (after constructing a sparse map that is diffused to a dense map via CNN layers). The SuperGlue network of Sarlin et al. is employed to compute the keypoints and their matching scores. The proposed approach is evaluated on two cardiac MRI segmentation datasets (ACDC and CHD) for which it shows superior performance compared to recent self-supervised approaches.



**Strengths:**

* The idea of using keypoints in a local contrastive loss for self-supervised pre-training is novel (to my knowledge) and interesting. Although it relies on the accurate detection and matching of keypoints, it has the potential to overcome the limitations of current strategies, for instance, based on spatial proximity and transformation consistency (which can not be used across different images).

* The method is sound and well presented.

* Experiments are detailed and results show clear improvements over recent self-supervised approaches for segmentation in few shot settings. Likewise, ablation studies and visualization experiments are nicely done and showcase the proposed method.

* The paper is quite well written.


**Weaknesses:**

* While the proposed method is compared against recent self-supervised approaches, only one of these approaches (GLCL) tries to learn a local representation, which makes the comparison a bit unfair. Yet, several approaches have been proposed for this purpose, for instance, see references below. As I understand, these approaches can also contrast local features that are semantically related (not necessarily spatially related), for example, based on clustering as in Peng et al, pseudo-labels as in Zhong et al., or super-pixels as in Chaitanya et al.

Wang, Zhaoqing, Qiang Li, Guoxin Zhang, Pengfei Wan, Wen Zheng, Nannan Wang, Mingming Gong, and Tongliang Liu. "Exploring set similarity for dense self-supervised representation learning." In Proceedings of the IEEE/CVF Conference on Computer Vision and Pattern Recognition, pp. 16590-16599. 2022.

Peng, Jizong, Marco Pedersoli, and Christian Desrosiers. "Boosting Semi-supervised Image Segmentation with Global and Local Mutual Information Regularization." Machine Learning for Biomedical Imaging 1, no. MIDL 2020 special issue (2021): 1-29.

Zhong, Yuanyi, Bodi Yuan, Hong Wu, Zhiqiang Yuan, Jian Peng, and Yu-Xiong Wang. "Pixel contrastive-consistent semi-supervised semantic segmentation." In Proceedings of the IEEE/CVF International Conference on Computer Vision, pp. 7273-7282. 2021.

Chaitanya, Krishna, Ertunc Erdil, Neerav Karani, and Ender Konukoglu. "Local contrastive loss with pseudo-label based self-training for semi-supervised medical image segmentation." Medical Image Analysis 87 (2023): 102792.

* Some of the reported results are not consistent with the literature. Namely, Table 1 reports a Dice of 0.642 for GLCL-full with M=2 in ACDC whereas the original paper reports a Dice of 0.789, which is higher than results of the proposed method.

* A potential weakness of the proposed method is that it requires to find matching local features across different images. Typically, descriptors like SIFT focus on points corresponding to sharp edges or blobs, which are present in cardiac structures. I suspect such points would be harder to find in structures with lower contrast such as prostate in MRI. Thus, it would be necessary to test the method on a broader range of organs and even other modalities like CT.



**Questions:**

* What is the threshold used to match keypoints?How sensitive is the method to the number of matching keypoints?

* Why are the results reported for GLCL-full lower than those of the original paper?

* Why not test the method on other segmentation tasks? For example, the GLCL paper by Chaitanya et al. also test on prostate segmentation.

* In the ablation study (Table 2), why not test the case where w1=w2=0? This would be useful since the main novelty of the method lies in L_local.

* In eq (2), does the index N+1 correspond to a no-match class?

Other comments:

* p5: "based on the cloest L2 distance" --> closest

**Limitations:**

Some limitations are mentioned in the discussion.

---

> ### Author Rebuttal · Authors · 2023-08-10
>
> > Comparison with other local SSL methods.
>
> Thank you for offering additional literature. We will add them in our paper.
>
> To clarify the difference between our method and `Peng et al., Zhong et al.`, `Chaitanya et al.`: while they incorporate local losses, their frameworks are designed for **semi**-supervised segmentation purpose, which requires a set of labeled dataset for joint training. Our method is designed for more general purpose **self**-supervised representation learning, which can be further finetuned on a small set of annotations. We agree that the local losses in these frameworks may be applicable for self-supervised training, but properly repurposing them may require further investigation and remain interesting future directions. `Wang el al.` is relevant to our approaches and they establish the local set correspondence between two different views of an input image. Unfortunately, we were unable to find its public source code, and reproducing the method (originally designed for natural images) on medical images requires additional consideration. We will include the comparison with `Wang et al.` if the authors open source their code in the future.
>
> Yet, for more comprehensive benchmarking purpose, we now add an additional *self-supervised* local SSL method as suggested by Reviewer `PaeV`: "CAiD: Context-Aware Instance Discrimination for Self-supervised Learning in Medical Imaging".
> Results are reported below. Our method outperforms CAiD in both datasets under different numbers of finetuning subjects (M). A full comparison will be attached into the paper.
>
> | | CHD | | ACDC | |
> |------------|:---:|------------|:----:|------------|
> | SSL Method | M=2 | M=15 | M=2 | M=6 |
> | *CAiD* | 0.265(.08) | 0.684(.04) | 0.483(.11) | 0.822(.02) |
> | Ours | **0.392(.06)** | **0.712(.03**) | **0.741(.03)** | **0.873(.01)** |
>
> >  Inconsistent Dice between GLCL-full and original paper
>
> Thank you for pointing this out! We believe this is a miscommunication on our part. In the original GLCL [1] paper, experiments are conducted based on a single fold train/test split, and results are reported on a held-out test set. In our experiment, we followed the more recent PCL [2]paper and conducted 5-fold cross validation. Table 1 indicates an average dice across the 5 folds. Note that our reported result of the GLCL-global is consistent with the GCL results reported in the PCL paper Table 1 (where they adopted the global loss of the GLCL). We will clarify the evaluation differences in our paper.
>
> [1] Chaitanya et al., "Contrastive learning of global and local features for medical image segmentation with limited annotations"
> [2] Zeng et al., "Positional Contrastive Learning for Volumetric Medical Image Segmentation"
>
> > Test the method on other organs and other modalities
>
> Thanks for your suggestion! We now add an additional multi-organ CT dataset Synapse [1], and the comparison with SoTA methods are below. Consistent with ACDC and CHD, our framework trained from randomly initialized weights or pretrained with SSL loss outperformed existing methods. We will supplement the full comparison in the paper.
>
> | Init. | Method | Dice (M=2) |
> |--------------|----------------------|------------|
> | Random Init. | Unet | 0.253(.06) |
> | | Swin-Unet | 0.198(.04) |
> | | SwinUNETR | 0.279(.06) |
> | | Ours | 0.289(.06) |
> | SSL pretrain | PCL | 0.306(.05) |
> | | Swin-Unet (with PCL) | 0.210(.07) |
> | | Ours | 0.322(.06) |
>
> [1] Synapse dataset: https://www.synapse.org/#!Synapse:syn3193805/wiki/217785
>
> > Threshold used to match keypoints and sensitivity
>
> Thank you. We will specify the threshold. We used 20 in the paper to find the correspondence. To verify how sensitive the method is to the number of matching keypoints, we perform an ablation on different threshold values, to pretrain our model on CHD, before finetuning it on $M=15$ labeled data. We turned off the global losses ($w_1=w_2=0$) to isolate the effects of the threshold for the local SSL loss. Average dice scores are reported below. As indicated from the results, our model remains robust across different values and the number of matching keypoints. We also include a visualization in the attached pdf with the number of matching keypoints annotated under different threshold values.
>
> | Threshold | Dice |
> |:---------:|:-------------:|
> | 5 | 0.689 (0.043) |
> | 10 | 0.690 (0.031) |
> | 15 | 0.684 (0.035) |
> | 20 | 0.689 (0.036) |
> | 25 | 0.684 (0.032) |
> | 30 | 0.685 (0.033) |
> | 35 | 0.690 (0.037) |
> | 40 | 0.687 (0.037) |
>
> > w1=w2=0 ablation
>
> Thank you. We have reported the results of $w_1=w_2=0$ and more ablations on the weight terms in our *appendix A. "Ablation on Correspondence Weights"*. Empirically, we find that only using the local loss for pretraining outperforms a randomly initialized UNet, which indicates that enforcing the localized representation across different slices offers a better initialization. However, a combined global and local loss during pretraining further increased the performance.
> This is also a consistent observation as in [1] and [2], where it is necessary that the local loss is combined with a global loss and/or other regularizations to avoid degenerated representations.
>
> [1] Ren et al., "Local Spatiotemporal Representation Learning for Longitudinally-consistent Neuroimage Analysis",
> [2] Chaitanya et al., "Contrastive learning of global and local features for medical image segmentation with limited annotations".
>
> > In eq (2), index N+1
>
> Yes. We follow [1] for the annotation. We will clarify this in the paper.
> [1] Sarlin et al., "SuperGlue: Learning Feature Matching with Graph Neural Networks"

---

> > ### Comment · Reviewer_b18J · 2023-08-11
> > **Thanks for the detailed answers**
> >
> > I have read the authors' answers carefully and I am generally satisfied with them. Regarding the first point (Comparison with other local SSL methods), I think the paper by Chaitanya et al. also evaluates their method in a self-supervised pre-training setup. Moreover, while it is true that the method Peng et al. is tested in a semi-supervised setup, it was extended to self-supervised learning in a follow up work:
> >
> > Peng, Jizong, Ping Wang, Marco Pedersoli, and Christian Desrosiers. "Boundary-aware information maximization for self-supervised medical image segmentation." arXiv preprint arXiv:2202.02371 (2022).

---

> > > ### Author Response · Authors · 2023-08-18
> > >
> > > Dear Reviewer b18J,
> > >
> > > Thank you very much for your positive comments, and for continuously pointing us toward the vital follow-up work of GLCL and Peng et al. For Peng et al., we were unable to find the open-source code yet, but we will add the reference in our manuscript. For Chaitanya et al., we conducted their semi-supervised training on ACDC dataset with M=2, and a comparison is reported as follows:
> > >
> > > | Dataset | Sample M | Method        | Mean/Std    |
> > > |---------|----------|---------------|-------------|
> > > | ACDC    | 2        | Local Semi by Chaitanya et al. [1] |  0.724(.071) |
> > > |         |          | Ours          | 0.741(.034) |
> > >
> > > We further acknowledge that self-supervised pretraining can be integrated with a semi-supervised framework, as pointed out by the reviewer. In Chaitaya et al., they first performed GLCL pretraining, which then serves as an initialization for the downstream joint semi-supervised finetuning. Similarly, our pretraining can serve as an alternative for initializing the network before performing downstream joint semi-supervised training.
> > >
> > > Please let us know if you have any further comments.
> > >
> > > [1]  Chaitanya, Krishna, Ertunc Erdil, Neerav Karani, and Ender Konukoglu. "Local contrastive loss with pseudo-label based self-training for semi-supervised medical image segmentation." Medical Image Analysis 87 (2023): 102792.

---

### Author Rebuttal · Authors · 2023-08-10


We thank the reviewers for their time and expert feedback.




As a summary of the reviews, we were happy to see that the submission was found to be well-written [`b18J,PaeV`], and it presents a sound and novel [`b18J,9qHo,m84B,8g4p,PaeV`] method for local self-supervised learning, with well formulated and detailed experiments [`b18J,PaeV`], which shows a significant advance [`b18J, m84B, 8g4p`].


Concerns raised included adding more related work [`b18J,PaeV`], comparisons with more local SSL methods [`b18J`] and transformer-based UNets [`9qHo`], experiments on non-cardiac datasets [`b18J, 8g4p`], ablations on key-point detection [`9qHo`], inconsistent GLCL results with the original paper [`b18J, 8g4p`], and scalability [`m84B,8g4p`].




To these ends, we address individual concerns below and will revise the paper accordingly. A few major responses are summarized as follows:




- As suggested by `b18J, PaeV`, we include one additional local SSL method (CAiD) to benchmark our method.
- As raised by `9qHo`, we now add comparison with SoTA Transformer-based unet architectures including Swin-Unet and Swin-UNETR as our benchmark.
- As mentioned by `b18J, 8g4p`, we now add results on an additional non-cardiac CT dataset (Synapse: the multi-organ segmentation benchmark).
- We clarified the difference between the GLCL results reported in the original paper (single fold train/validation/test) and ours (5-fold cross validation).
- As requested by `9qHo` and `b18J`, we provided more ablation analysis on key-point detection methods, as well as various thresholds for determining the correspondence, to evaluate the sensitivity to the keypoint detection and matching. We also attached an additional figure to visualize the detected correspondence.
- We revised Figure 1 and attached the Dice score suggested by `m84B`.
- Typos, missing technical details, and related works will be revised in the paper.




Again, we deeply appreciate the feedback and are happy to receive any further questions, comments, and/or suggestions for improvements.

---

### Decision · Program_Chairs · 2023-09-21

**Decision:**

Accept (poster)

**Comment:**

The paper introduces a new self-supervised pre-training mechanism based on keypoints to match spatially distant features for medical image segmentation. The method shows superior results compared to other self-supervised methods on two cardiac MRI segmentation benchmarks (ACDC and CHD). The paper received scores 5,6,6,6,3. The main strengths of the paper are: 1) novelty of using keypoints for self-supervised pre-training; 2) clarity; and 3) thorough experiments. The main weaknesses are: 1) unfair comparison with global representation approaches; 2) inconsistent results with literature; 3) potential disadvantage of the need to detect keypoints; 4) lack of comparison with recent UNet-based methods; 5) lack of complexity analysis; 6) challenging reproducibility; and 7) lack of significance tests. There was a lot of discussion about the paper and rebuttal, where many of issues raised by the reviewers were addressed. The authors should take into account these discussion points when preparing the final version of the paper.